

# The Geometric Correction Method for zircon (U-Th)/He chronology: correcting systematic error and assigning uncertainties to alpha-ejection corrections and eU concentrations

Spencer D. Zeigler[1,*], Morgan Baker[1,*], James R. Metcalf[1], Rebecca M. Flowers[1]

[1]Department of Geological Sciences, University of Colorado Boulder, Boulder, CO 80309, USA
*These authors contributed equally to this work.

*Correspondence to*: Spencer D. Zeigler (spencer.zeigler@colorado.com)

**Abstract.** The conventional zircon (U-Th)/He (ZHe) method typically uses microscopy measurements
of the dated grain together with the assumption that the zircon can be appropriately modeled as a
geometrically perfect tetragonal or ellipsoidal prism in the calculation of volume (V), alpha ejection
correction ($F_T$), equivalent spherical radius ($R_{FT}$), effective uranium concentration (eU), and corrected
(U-Th)/He date. Here, we develop a set of corrections for systematic error and determine uncertainties
to be used in the calculation of the above parameters for zircon, using the same methodology as Zeigler
et al. (2023) for apatite. Our approach involved acquiring both "2D" microscopy measurements and
high resolution "3D" nano-computed tomography (CT) data for a suite of 223 zircon grains from nine
samples showcasing a wide range of morphology, size, age, and lithological source, calculating the V,
$F_T$, and $R_{FT}$ values for the 2D and 3D measurements, and comparing the 2D vs. 3D results. We find that
the values derived from the 2D microscopy data overestimate the true 3D V, $F_T$, and $R_{FT}$ values for
zircon, with one exception (V of ellipsoidal grains). Correction factors for this misestimation
determined by regressing the 3D vs. 2D data range from 0.81-1.04 for V, 0.97-1.0 for $F_T$, and 0.92-0.98
for $R_{FT}$, depending on zircon geometry. Uncertainties (1σ) derived from the scatter of data around the
regression line are 13-21% for V, 5%-1% for $F_T$, and 8% for $R_{FT}$, again depending on zircon
morphologies. Like for apatite, the main control on the magnitude of the corrections and uncertainties is
grain geometry, with grain size being a secondary control on $F_T$ uncertainty. Propagating these
uncertainties into a real dataset (N = 28 ZHe analyses) generates 1σ uncertainties of 12-21% in eU and
3-7% in the corrected ZHe date when both analytical and geometric uncertainties are included.
Accounting for the geometric corrections and uncertainties is important for appropriately reporting,
plotting, and interpreting ZHe data. For both zircon and apatite, the Geometric Correction Method is a
practical and straightforward approach for calculating more accurate (U-Th)/He data, and for including
geometric uncertainty in eU and date uncertainties.

## 1 Introduction

The conventional whole-crystal technique for zircon (U-Th)/He (ZHe) geo- and thermochronology is
used for a variety of studies including precisely dating volcanic eruptions (e.g., Danišík et al., 2021),





constraining the timing of tectonic exhumation (e.g., Reiners et al., 2002), deciphering sedimentary provenance (e.g., Stockli and Najman, 2020), and inferring erosion associated with unconformity development (e.g., Orme et al., 2016; Flowers et al., 2020). The ZHe method requires microscopy measurements of the zircon length and widths as well as measurements of parent and daughter amounts. The microscopy measurements are typically used in conjunction with an idealized geometric model of a
tetragonal or ellipsoidal prism (e.g., Ketcham et al., 2011; Figure 1) to calculate the zircon's volume (V) and surface area. These geometric parameters are then used to compute the alpha-ejection correction ($F_T$), the effective uranium concentration (eU), and the equivalent spherical radius. $F_T$ values are necessary to correct ZHe dates for the $^4$He atoms ejected from the crystal lattice during decay (e.g., Farley et al., 1996; Ketcham et al., 2011). eU (a proxy for radiation damage) is a critical parameter for
interpreting ZHe dates because the retentivity of $^4$He is a function of radiation damage (e.g., Guenthner et al., 2013; Ginster et al., 2019). The equivalent spherical radius approximates the diffusion domain of a whole crystal and is needed for thermal history modeling (here, we use $R_{FT}$, the radius of a sphere with an $F_T$ correction the same as the analyzed grain).

Variations in zircon morphology and termination shape can cause real grains to deviate from the perfect geometric prisms assumed by the microscopy method for computing the geometric parameters and associated values (Figure 1), causing both uncertainty and possibly systematic error in these data. Here, "uncertainty" refers to the measurement reproducibility (i.e., the precision), while "error" refers to the systematic deviation between a measured value and the true value (i.e., the accuracy) (JCGM, 2012).
Quantifying the uncertainties and systematic error arising from the use of an idealized geometry to calculate geometric parameters is needed to assign appropriate uncertainties to ZHe data and to derive accurate results.

For the mineral apatite, previous work has focused on characterizing and reducing uncertainties and
systematic error on the geometric parameters using X-ray micro- or nano- computed tomography (CT) (Herman et al., 2007; Evans et al., 2008; Glotzbach et al., 2019; Cooperdock et al., 2019; Zeigler et al., 2023). CT is a high resolution, non-destructive method that creates 3D models of scanned objects from which high quality 3D geometric data such as volume and surface area can be extracted using software like Blob3D (Ketcham, 2005). While CT data have collectively been acquired for several hundred
apatite grains over the course of several studies (Herman et al., 2007; Evans et al., 2008; Glotzbach et al., 2019; Cooperdock et al., 2019; Zeigler et al., 2023), only 5 zircon grains have been analyzed by CT (Evans et al., 2008) and no study has comprehensively addressed uncertainty and error in the zircon geometric parameters.

Here, we fill this gap by 1) presenting high resolution CT data for 223 zircon grains representative of a wide range of morphology, size, age, and lithologic source and 2) developing a zircon "geometric correction method" to regularly correct for systematic error and to assign uncertainties for zircon V, $F_T$, and $R_{FT}$ that can be propagated into the eU value and ZHe date. This study follows the approach of Zeigler et al. (2023) for apatite and generates a method that similarly involves no added work or cost
beyond what is already done as part of most existing (U–Th) / He dating workflows, and that can be applied retroactively to previously collected data. Like in Zeigler et al. (2023), we first developed a





"grain evaluation matrix" (GEM) for zircon that classifies grains based on their morphology, acquired "2D" microscopy measurements and high resolution (0.84-0.92 μm) "3D" CT data of the same zircon grains, compared the grain dimension measurements, V, $F_T$, and $R_{FT}$ values, regressed the 3D vs. 2D
data, and then determined a set of corrections and uncertainties based on grain geometry and size. While in the past geometric parameters have not been corrected for systematic error and uncertainties in a zircon's geometric information have not typically been propagated into eU and ZHe date uncertainties, the Geometric Correction Method provides a straightforward approach for addressing both of these issues. We illustrate the method with real ZHe data to show its importance for the accuracy and
precision of ZHe datasets.

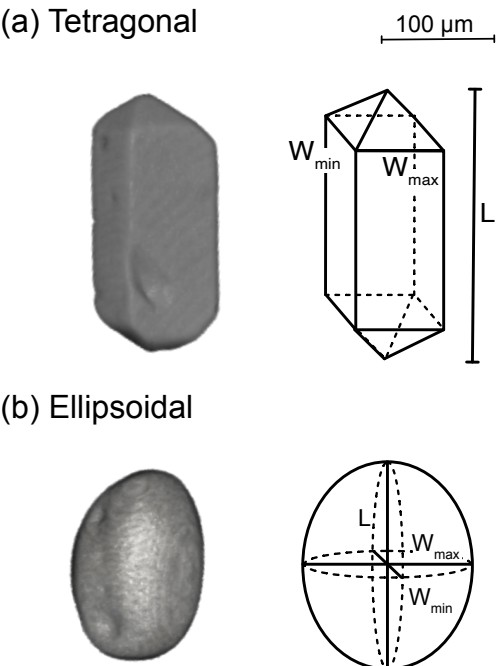

**Figure 1. 3D renderings from CT data of real zircon crystals classified as (a) tetragonal and (b) ellipsoidal versus the idealized geometry from Ketcham et al. (2011) that is used to calculate V, $F_T$, and $R_{FT}$. Scale bar is applicable to both examples of real**
**crystals. Note that the actual grains have geometries that are not perfectly represented by the idealized geometry. The grain length (L), maximum width ($W_{max}$), and minimum width ($W_{min}$) denoted on the schematics of the idealized geometries represent the three grain measurements made using standard 2D microscopy measurements in this study.**





**Table 1. Zircon sample information.**

| Sample Name | Unit and Lithology | Sample Age | Locality | Latitude (°N) | Longitude (°W) | GEM Categories | N[a] | Additional Geochronologic and Thermochronologic Data |
|---|---|---|---|---|---|---|---|---|
| FCT | Fish Canyon Tuff, Dacite | Oligocene | San Juan Mountains, Colorado, USA | 37.756 | 106.934 | A1, A2 | 23 | Zircon U-Pb 28.172 ± 0.028 Ma (2σ) (Schmitz and Bowring, 2001); ZHe 28.7 ± 0.4 Ma (1σ) (Gleadow et al., 2015) |
| RGD17-21 | Harrison Pass Pluton, Granodiorite | Eocene | Ruby Mountains, Nevada, USA | 40.326 | 115.510 | A1, A2, B1, B2, B3 | 23 | Zircon U-Pb ca. 36 Ma (Wright and Snoke, 1993); ZHe 20-16 Ma[b] (McGrew & Metcalf., 2000) |
| CA8 | Potomac terrane, Quartzofeldspathic schist | Precambrian | Appalachian Mountains, Virginia, USA | 37.984 | 78.311 | A1, A2, B1, B2 | 27 | ZHe 186-121 Ma[b] (Basler et al., 2021) |
| PP4 | Pikes Peak Batholith, Syenogranite | Proterozoic | Pikes Peak, Colorado, USA | 38.842 | 105.025 | A1, A2, A3, B2 | 20 | Hornblende & Biotite 40Ar/39Ar 1.08-1.07 Ga (Unruh, 1995); ZHe 115- 773 Ma[b] (Havranek and Flowers, 2022) |
| CP06-70 | 245-Mile Complex, Granodiorite | Proterozoic | Grand Canyon, Arizona, USA | 35.843 | 113.599 | A3, B3, C2, C3, B1, A2, A1 | 39 | Zircon U-Pb ca. 1700 Ma (Hawkins et al., 1996); ZHe 560-96 Ma[b] (Peak et al., 2021) |
| 01-OE-38 | Migmatitic Gneiss | Archean | Superior craton, Canada | 47.270 | 84.560 | A2, A3, B2, B3 | 24 | Zircon U–Pb 2720–2680 Ma (Hoffman, 1989); AHe 275-34 Ma[b] (TRaIL unpublished data) |
| 56JBM14 | Río de los Patos Frm., tuffaceous sandstone | Paleogene | Manantiales Basin, Argentina | -32.050 | 69.750 | B1, B2, B3 | 10 | Zircon U-Pb 38.68 ± 0.21 (2σ) (Suriano et al., 2023) |
| CP06-14 | Coconino Sandstone | Permian | Colorado Plateau, Arizona, USA | 34.300 | 110.901 | C1, C2, C3 | 28 | No geochronologic data for this sample |
| CP06-15 | Esplanade Sandstone | Permian | Colorado Plateau, Arizona, USA | 34.298 | 110.906 | B3, C1, C2, C3, B2 | 29 | No geochronologic data for this sample |

**[a] The number of grains for which high quality CT data were acquired.**

**[b] Range of single grain ZHe dates from this sample.**

## 2 Selecting and characterizing a representative zircon suite

### 2.1 Strategy

In this study we selected zircon grains reflecting the full spectrum of zircon characteristics so that the outcomes are applicable to the range of grains commonly dated by (U-Th)/He. As described in more detail below, we focused on choosing zircons from a variety of source lithologies and ages (Sect. 2.2),
with a range of grain sizes (Sect. 2.3) and morphologies (including grain geometry, number of terminations, and radiation damage) (Sect. 2.4). We originally selected 326 grains for CT analysis and ended up with 223 grains with high-quality CT data.

### 2.2 Selecting a representative zircon sample suite

The zircon sample suite contains six igneous and metamorphic rocks and three sedimentary samples
(Table 1). All samples were separated using standard crushing, density, and magnetic separation techniques. Five of the nine samples were dated previously by ZHe in the CU TRaIL (Thermochronology Research and Instrumentation Lab). The Oligocene Fish Canyon Tuff (sample FCT) has ZHe dates overlapping emplacement (e.g., Dobson et al., 2008; Gleadow et al., 2015). The Eocene Harrison Pass pluton (sample RGD17-21) from the southern Ruby Mountains of Nevada yields
Miocene ZHe dates (McGrew and Metcalf, 2020). Three Proterozoic samples include a Neoproterozoic



quartzofeldspathic schist from the central Appalachians (sample CA8) that yields Mesozoic ZHe dates (Basler et al., 2021), the ~1.1 Ga Pikes Peak granite from Pikes Peak, Colorado (sample PP4) with a span of Cryogenian and younger ZHe dates (Havranek and Flowers, 2022), and a Proterozoic granodiorite from the 245 Mile Pluton in the Lower Granite Gorge of the Grand Canyon in Arizona,
USA (sample CP06-70) that yields Ediacaran and younger ZHe dates (Peak et al., 2021). Zircon grains from an Archean migmatitic gneiss sample (01-OE-38) from the Superior craton in Canada were also included in this study. The three detrital samples include a Neogene sedimentary unit (sample 56JBM14) from the Manantiales Basin in Argentina, as well as samples of the Permian Coconino Sandstone (CP06-14) and the Permian Esplanade Sandstone (sample CP06-15) from the Colorado
Plateau in northeastern Arizona, USA.

**2.3 Selecting a representative zircon crystal size distribution**

The size distribution of grains analyzed in this study is based on the size distribution of grains routinely analyzed for (U-Th)/He dates. We first plotted the maximum width of all zircon (N = 736; Fig. 2) analyzed in the CU TraIL over a two year period. The grains in this compiled dataset were from a
variety of sources and were selected and measured by TRaIL staff, TRaIL students, and visitors. Our analysis focused on crystal width because the smallest dimension (i.e., the width) is the chief control on alpha ejection due to the long stopping distances of alpha particles. These lab analyses were subdivided into small (< 50 µm maximum width), medium (50-100 µm maximum width), and large (>100µm maximum width) size categories (shading in Fig. 2). We based our size categories on the maximum
width only for consistency with our complementary apatite study (Zeigler et al., 2023). From the samples described above we then picked suites of zircon crystals for CT with size distributions that closely matched that in the compiled datasets (Fig. 2). For zircon, the grains in our final dataset range in maximum width from 34 µm to 153 µm.



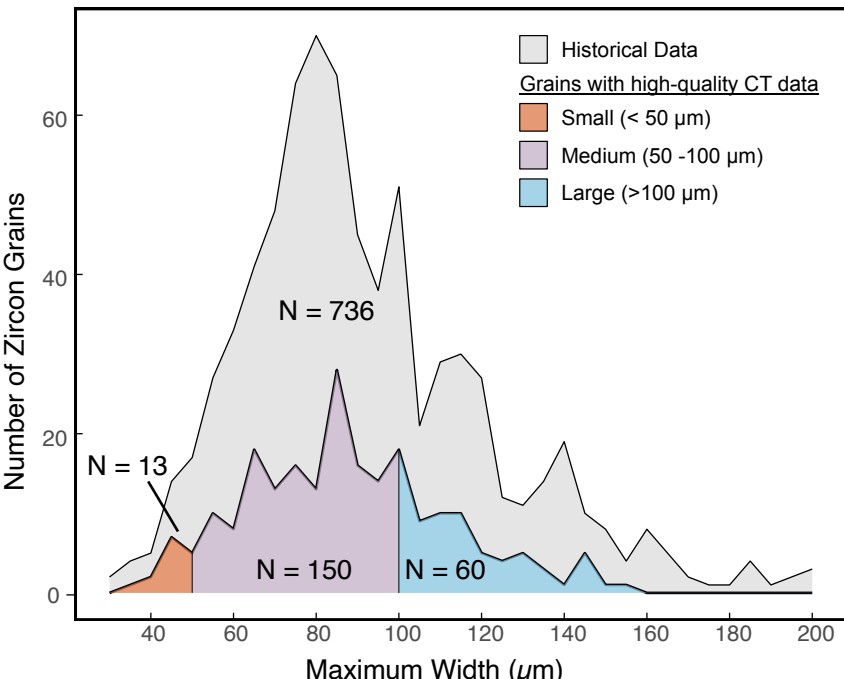

**Figure 2. The distribution of maximum widths of zircon in this study. Light grey depicts 736 zircon grains dated in the CU TRaIL between 2017-2019. Colored shading illustrates the size distribution of all grains for which we acquired high-quality CT data, with the number of grains in each size category listed.**

## 2.4 Selecting a morphologically representative zircon crystal suite and designing the Zircon Grain Evaluation Matrix

To select a representative zircon sample suite in terms of morphology, first we carefully inspected each sample and noted the variety and proportions of zircon morphological characteristics. In addition to grain shape (euhedral vs. rounded), we noted the grain color, the grain clarity, and the number and shape of the terminations. Termination shapes included pointed terminations like those in the tetrahedral prism of Ketcham et al. (2011), "taco" terminations where the points of the terminations are not centered over the trunk of the grain and are instead off to one side, and "hipped roof" terminations where the terminations end in a ridge rather than a point. Then, when picking the zircon grains for CT analysis, we ensured that the variety of grain morphologies was accurately reflected and that similar percentages of grains with 0, 1, or 2 terminations were included as are in the compiled TRaIL zircon dataset (Sect 2.3).

We used this initial survey of our samples to develop a zircon Grain Evaluation Matrix (GEM), much like that for apatite (Zeigler et al., 2023). The zircon-GEM was initially designed with two axes: a "geometric classification" x-axis and a "clarity index" y-axis (see Appendix B, Fig. B1). Grain clarity was initially considered because this characteristic correlates with radiation damage (e.g., Ault et al., 2018), which influences zircon He retentivity (and therefore the ZHe date) and zircon density (and therefore the estimated mass and eU values). Grain clarity thus can be useful information to record





during grain selection and is retained in the two-axis zircon GEM in the appendix. However, zircon
clarity does not impact geometric corrections and uncertainties (Table C2), so the zircon GEM was
collapsed to a single "geometric classification" axis (Fig. 3). The geometry is described as A
(tetragonal), B (sub-tetragonal), or C (ellipsoidal). A and B grains assume a tetragonal geometry while
C grains assume an ellipsoidal geometry for 2D geometric parameter calculations (see Sect. 3.2,
Appendix A).

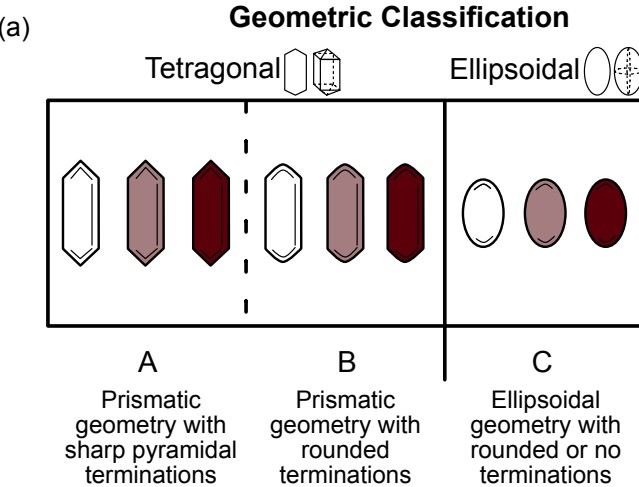

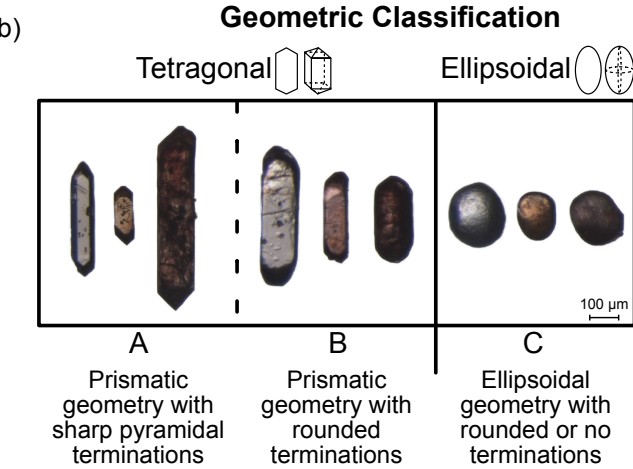

**Figure 3. The zircon Grain Evaluation Matrix (GEM) in (a) schematic form and (b) with images of real grains analyzed in this**
**study. The geometric axis classifies grains as A, B, or C, where both A and B zircon grains assume an idealized tetragonal prism**
**geometry while C zircon grains assume an idealized ellipsoidal geometry for 2D calculations (Ketcham et al., 2011).**



# 3 Measurement and data reduction methods

## 3.1 Strategy

The goal of this work is to determine corrections for systematic error and appropriate uncertainties arising from traditional "2D" microscopy measurements by comparing the 2D geometric parameters with "3D" geometric parameters acquired via CT. To accomplish this, we first measured our representative sample suite using the conventional 2D microscopy approach (Sect 3.2) and acquired high resolution (0.84-0.92 µm) 3D CT data for those same grains (Sect. 3.3). Then, we examined the relationship between 2D and 3D measurements, used linear regression to determine the corrections

based on grain geometry (tetragonal vs. ellipsoidal), and calculated the uncertainty (Sect. 3.4). This analysis assumes that the 3D CT measurements are accurate (Sect. 3.3). $F_T$ uncertainties include only those uncertainties associated with grain geometry and not those due to parent isotope zonation, grain abrasion, or crystal breakage.

## 3.2 Microscopy measurements and 2D calculation methods

Zircon grains were hand-picked under a Leica M165 binocular microscope under 160× magnification. Each grain was photographed on a Leica DMC5400 digital camera, manually measured using either the Leica LAS X or Leica LAS 4.12 software, and assigned a GEM value (Fig. 3). The calibration of the software was checked before, during, and after the measurements using a micrometer. Photomicrographs were taken under plane-polarized light with the c-axis in the same orientation to

properly assess the color and clarity of the grain. The 2D measurement procedure for zircon is shown in Figure 4. First, the length was measured parallel to the c-axis and the maximum width was measured perpendicular to the c-axis. Then, the grain was rotated 90° onto its side to acquire a second length (parallel to the c-axis) and a minimum width (perpendicular to the c-axis). The flat-sided habit of zircon makes it straightforward to measure both widths and the grain length accurately, so we used both widths

and the average of the two length measurements for the 2D calculations.



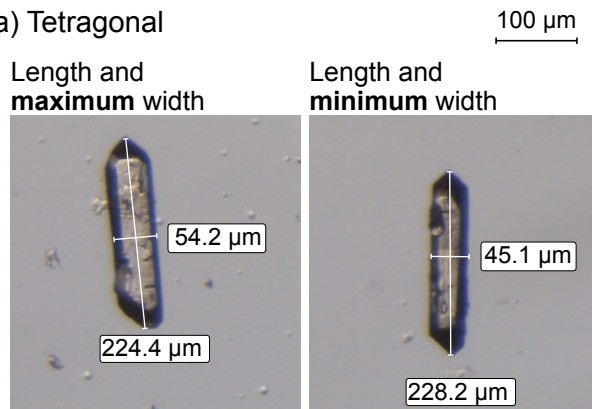

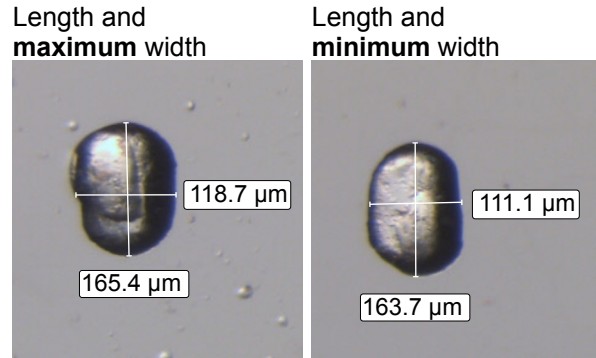

**Figure 4. Photomicrographs of (a) tetragonal and (b) ellipsoidal zircon grains showing how each grain's (average) length, maximum width, and minimum width were measured for the 2D microscopy measurements. After measuring the first length and maximum width, the grain was rolled 90° onto its side, another photomicrograph of the grain was acquired, and a second length and the minimum width were measured.**

The 2D V and isotope-specific $F_T$ values were calculated using the equations of Ketcham et al. (2011) and assuming a tetragonal prism geometry for all GEM A and B grains and an ellipsoidal geometry for all GEM C grains. $R_{FT}$ calculations use the equations from Cooperdock et al., 2019. Appendix A lists all equations. We used the mean alpha stopping distances for $^{238}$U, $^{235}$U, $^{232}$Th, and $^{147}$Sm from Ketcham et al. (2011). The $F_T$ calculations of Ketcham et al. (2011) assume that every surface is an ejection surface. For each zircon, we calculated the combined $F_T$ value, and the associated $R_{FT}$ value, by assuming a zircon Th/U ratio of 0.87 and no Sm contribution, based on the average of the TRaIL zircon sample historical data (N = 736 grains) shown in Figure 2. We made this assumption because the combined $F_T$ and $R_{FT}$ values depend on the proportion of each parent isotope contributing to $^4$He production, and we do not have parent isotope values for the grains analyzed by CT in this study.



### 3.3 Nano-computed tomography (CT) and 3D calculation methods

After 2D measurements, zircon grains were mounted for CT. Zircon crystals were mounted in an ~1500
x 1500 μm area on a thin, 2000 μm wide plastic disc that was hole-punched from a plastic sheet
protector and then covered with double sided tape (Fig. 5). Each plastic disc held 4-10 grains and 5-6
discs were stacked vertically to create a mount (Fig. 5). Mounts were secured by a thin layer of
parafilm, attached to a 1-2 mm thick cylinder of rubber for stabilization, and then glued to the head of a
flat-head pin (Fig. 5).

Each mount was scanned on a Zeiss Xradia 520 Versa X-ray Microscope in the University of Colorado
Boulder Materials Instrumentation and Multimodal Imaging Core (MIMIC) Facility. Scanning
parameters were optimized to reduce noise and scanning artifacts during test scans of the first mount.
Scanning parameters were held relatively constant for subsequent mounts with minor adjustments to
optimize the tradeoff between scan time and resolution (Table B1). All mounts were scanned with the
4X objective at high power and voltages, which allowed for high resolution (0.84-0.92 μm).

Raw CT data was processed in Blob3D following the methods outlined in Zeigler et al. (2023). 3D
parameters such as grain dimensions (Box A, B, C), V, and isotope specific $F_T$ values were extracted
from Blob3D (see Sect. 4.3 in Zeigler et al., 2023). Like the calculations done for 2D $R_{FT}$ values, we
calculated 3D $R_{FT}$ values using the equations of Cooperdock et al. (2019) and assuming a Th/U ratio of
0.87 based on TRaIL zircon sample historical data. We assume that the CT measurements are
representative of the "real" grain measurements because previous work showed that the ± 1 %
uncertainty of the CT measurements translates to negligible differences in the relevant values output by
Blob3D (Cooperdock et al., 2019; Zeigler et al., 2023).

Some zircon grains were removed from the final dataset owing to issues during CT scanning or
subsequent data processing. The entirety of zircon mount 2 was excluded due to analytical problems
during CT scanning. Three additional zircon grains were excluded owing to 3D models that presented
large holes or had many small gaps that caused the model to be a hollow shell, possibly due to less
dense inclusions at the grain edge. The final dataset consists of 223 crystals out of the initial set of 326
grains.



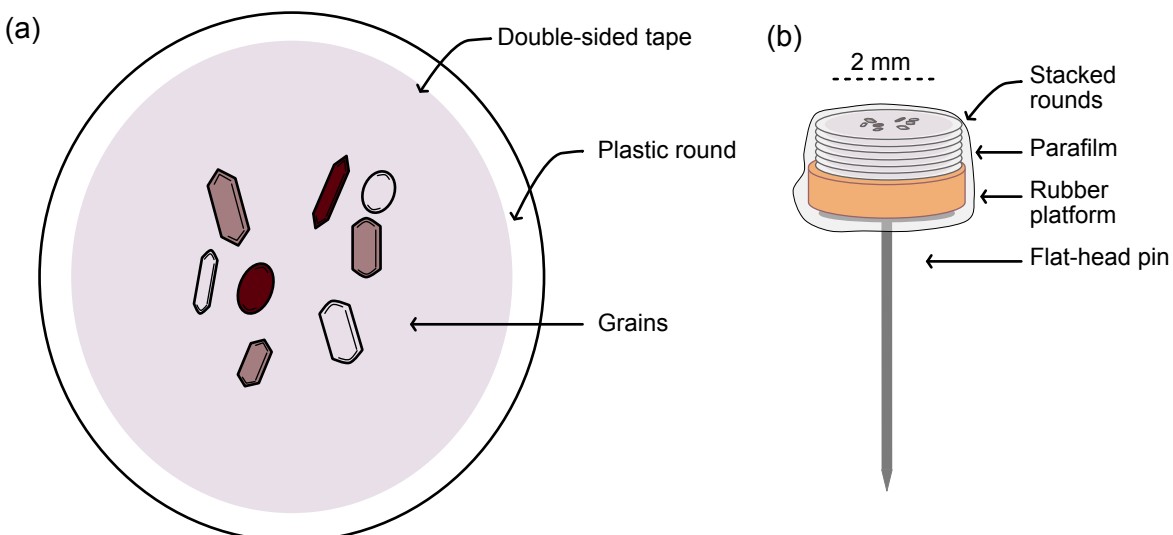

**Figure 5. Schematic showing (a) an individual plastic round and (b) a final grain mount for CT analysis. Grains are placed onto a ~2 mm wide sturdy plastic disc (hole punched from a plastic sheet protector) covered with double-sided tape. Each plastic round can hold between 4 and 10 grains. Rounds are stacked on top of each other and placed on a rubber platform cut from old test tube stoppers, which is glued to a flathead pin and covered with Parafilm.**

### 3.4 Statistical comparison of 2D and 3D values

The first step in our statistical analysis was to create scatter plots of the 3D vs. 2D values for the grain dimensions (Fig. 6), V, $F_T$, and $R_{FT}$ (Fig. 7). In Figure 7, we only show the results for $^{238}F_T$ because it dominates the $^4$He budget, but the results for each isotope ($^{235}$U, $^{232}$Th, and $^{147}$Sm) are in Appendix Figure C1. For completeness we include the $^{147}F_T$ value, but $^{147}$Sm is commonly not measured in zircon because of its negligible incorporation into this mineral and insignificant contribution to the helium budget. Although surface area is a parameter output by Blob3D, we did not consider it separately because surface area alone is not used to calculate any parameters.

We next regressed the 3D vs. 2D data for each parameter. If the data plot on the 1:1 line (bold black line) on the 3D versus 2D plots, then no correction for systematic error is needed for the 2D data because the 2D and 3D data agree. However, if the data plot above or below the 1:1 line, then the correction desired for the 2D data can be viewed as the offset of the data and its linear regression line from the 1:1 line. To determine the corrections for systematic error we followed the procedure outlined in Sect. 4.4 of Zeigler et al (2023) and used a simple, ordinary least squares linear regression with the intercept fixed at 0.

We tested different groupings of physical parameters for the linear regressions to assess which groups yielded statistically different slopes (i.e., corrections for systematic error) using Tukey's Highly Significant Difference test. The results of this analysis are in Table C1. We found that the slopes were statistically indistinguishable when regressions were grouped by size (medium and small (<100μm) vs. large (>100μm)) or clarity (1, 2, or 3; Fig. B1), but a significant difference was found between the



275 slopes for GEM A and B (tetragonal) and GEM C (ellipsoidal) zircon grains. Therefore, the regressions are grouped by grain geometry for all geometric parameters.

The uncertainty for V, $F_T$, and $R_{FT}$ is the scatter of the data around the regression line. To quantify the uncertainty, we used the approach of Zeigler et al. (2023) and computed the $1\sigma$ standard deviation of the
280 residual values of the points from the regression line, plotted as residual % difference versus maximum width for each parameter (Fig. 7d-f). The standard deviations for different groups of physical variables (e.g., size, clarity) were compared to evaluate potential relationships with uncertainty (Table C2). We used Pearson's r to calculate the correlation coefficient between each isotope-specific $F_T$ uncertainty (Martin et al., 2023).
285

**Figure 6. Scatter plots of 3D vs. 2D data (N = 223) for grain dimension measurements. (a) 3D Box A vs. 2D length measurement, (b) 3D Box B vs. 2D maximum width measurement, and (c) 3D Box C vs. 2D minimum width measurement. The bold black line is the 1:1 line.**









**Figure 7. Plots illustrating how the corrections for systematic error and how uncertainties were determined for V, $F_T$, and $R_{FT}$. Scatter plots of 3D vs. 2D data (N = 223) with regression lines and data distinguished by geometry for (a) V, (b) $^{238}F_T$, and (c) $R_{FT}$. The bold black line is the 1:1 line and the dashed lines mark the percent difference from the 1:1 line. Note that for all regressions (except for the volume and $F_T$ of ellipsoidal grains), the regression line falls below the 1:1 line, indicating that the 2D-microscopy data overestimate the 3D-CT data. The 2D data can be corrected for systematic error by multiplying the 2D data by the slope of the regression line. Plots of the difference of each 2D value from the regression line (i.e., the residual) as a percent difference vs. maximum width with data distinguished by geometry for (d) V, (e) $^{238}F_T$, and (f) $R_{FT}$. For $^{238}F_T$ the tetragonal grains are additionally split by <100 µm maximum width (small- and medium-sized grains of Fig. 2) and > 100 µm maximum width (large-sized grains of Fig. 2). The bold black line is 0% difference. Note the larger y-axis scale for V as compared with $^{238}F_T$ and $R_{FT}$, reflecting the greater uncertainty of V. The standard deviation of the % difference in the residuals of each group is the uncertainty on the parameter.**

**Table 2. Corrections and uncertainties (1σ) for all geometric parameters.**

**Volume**

| Geometry | Correction[a] | % Uncert.[b] (1σ) for zircon grains of all sizes |
|---|---|---|
| Tet. | 0.81 | 13% |
| Ellip. | 1.04 | 21% |

**Isotope-specific $F_T$ values**

| Geometry | Correction | % Uncert. (1σ) for zircon grains with $w_{max}$ <100 µm | % Uncert. (1σ) for zircon grains with $w_{max}$ >100 µm |
|---|---|---|---|
| $^{238}F_T$ | | | |
| Tet. | 0.97 | 3% | 2% |
| Ellip. | 1 | 3% | 3% |
| $^{235}F_T$ | | | |
| Tet. | 0.97 | 4% | 3% |
| Ellip. | 1 | 4% | 4% |
| $^{232}F_T$ | | | |
| Tet. | 0.97 | 5% | 3% |
| Ellip. | 1 | 4% | 4% |
| $^{147}F_T$ | | | |
| Tet. | 0.99 | 1% | 1% |
| Ellip. | 1 | 1% | 1% |

| $R_{FT}$ | | |
|---|---|---|
| Geometry | Correction | % Uncert. (1σ) for zircon grains of all sizes |
| Tet. | 0.92 | 8% |
| Ellip. | 0.98 | 8% |

[a] **The correction value is the slope of the 3D vs. 2D regression line for each parameter in Figures 7a-7c and Figure C1a-c.**

[b] **The uncertainty is the scatter of the 2D data about each regression line in Figure 7a-c and Figure C1a-c, calculated as the 1σ standard deviation of the % difference of each 2D value from the regression line (Fig. 7d-f and Fig. C1-d-f).**



**4 Results**

For zircon dimension data, the 3D values closely match the 2D values for length (Box A), maximum width (Box B), and minimum width (Box C) (Fig. 6a-c). The average 3D/2D value and average absolute percent differences are 0.99 and 3% for length/Box A, 1.04 and 6% for maximum width/Box B, and

1.02 and 8% for minimum width/Box C (Table 3). 2D measurements are generally accurate for zircon, owing partially to its rectangular habit which makes 2D measurements relatively straightforward. Outliers on Figure 6 can be attributed to grains with uneven terminations, sharp broken angles, or otherwise unusual morphologies which can cause Blob3D to measure dimensions different from the c-axis parallel length and perpendicular widths used in the 2D measurements (Cooperdock et al., 2019).


The 3D vs. 2D scatter plots for zircon V, $F_T$, and $R_{FT}$ show data that systematically plot on or below the 1:1 line (bold black line) with one exception (V of ellipsoidal grains), indicating that for almost all parameters, the 2D values overestimate the true 3D values. The 2D data can be corrected for their offset from the 3D data by multiplying the 2D data by the slope of the 3D vs. 2D data so that the 2D data are

centered around the 1:1 line, thereby "correcting" them. As noted in Sect. 3.4, regressions of the 3D vs. 2D data are separated by geometry because the regressions of tetragonal and ellipsoidal grains yield statistically distinguishable slopes. The corrections for systematic error for zircon V, $F_T$, and $R_{FT}$ are summarized in Table 2. For all parameters, the magnitude of the correction is larger for tetragonal grains than for ellipsoidal grains (Table 2; (e.g., for $F_T$, a 0.97 correction for tetragonal grains vs. no

correction for ellipsoidal grains)).

The uncertainties for V, $F_T$, and $R_{FT}$ are derived from the scatterplots of percent difference in the residuals versus maximum width in Figure 7d-f, where the bold black line represents a 0% difference between the 2D and 3D data. The uncertainties are separated by geometry for all parameters because the

residuals are derived from the regression lines, which group the data in this way (Table 2). A single uncertainty is reported for ellipsoidal zircons for all parameters due to the relatively small number of ellipsoidal grains in the dataset (N = 61). However, because the size of the tetragonal dataset was large (N = 162), we explored different uncertainty groupings (Table C2). For V and $R_{FT}$, neither grain size nor clarity had a consistent or clear relationship with uncertainty (Table C2). However, for $F_T$, grain size

and uncertainty are related, with larger uncertainty for smaller grain size. For all parameters, the uncertainty for tetragonal grains is smaller than or equal to the uncertainty for ellipsoidal grains (Table 2). For example, for V, the uncertainties for tetragonal and ellipsoidal grains are 13% and 21%, respectively. As anticipated, the isotope-specific $F_T$ uncertainties are correlated, yielding correlation coefficients of 0.79-0.99.





**Table 3. 2D microscopy and 3D CT data comparison for this study[a]**

| This Study: 223 zircon grains; avg. CT resolution: 0.87 µm | | | | |
|---|---|---|---|---|
| | Avg. 3D/2D[b] | 1σ | Abs. avg. % diff.[c] | 1σ |
| **All data: 223 grains** | | | | |
| Volume | 0.88 | 0.19 | 19% | 12 |
| $^{238}F_T$ | 0.98 | 0.03 | 3% | 3 |
| $R_{FT}$ | 0.94 | 0.08 | 8% | 6 |
| Length/Box A | 0.99 | 0.05 | 3% | 3 |
| $W_{max}$/Box B | 1.04 | 0.09 | 6% | 7 |
| $W_{min}$/Box C | 1.02 | 0.1 | 8% | 7 |
| **Tetragonal zircon: 162 grains** | | | | |
| Volume | 0.81 | 0.1 | 20% | 10 |
| $^{238}F_T$ | 0.97 | 0.03 | 3% | 3 |
| $R_{FT}$ | 0.92 | 0.07 | 9% | 6 |
| Length/Box A | 0.99 | 0.04 | 3% | 3 |
| $W_{max}$/Box B | 1.03 | 0.09 | 7% | 7 |
| $W_{min}$/Box C | 1.03 | 0.09 | 7% | 7 |
| **Ellipsoidal zircon: 61 grains** | | | | |
| Volume | 1.09 | 0.22 | 18% | 16 |
| $^{238}F_T$ | 1 | 0.03 | 2% | 2 |
| $R_{FT}$ | 0.99 | 0.08 | 7% | 4 |
| Length/Box A | 1.02 | 0.05 | 3% | 4 |
| $W_{max}$/Box B | 1.04 | 0.07 | 6% | 6 |
| $W_{min}$/Box C | 0.98 | 0.12 | 9% | 8 |

**[a] Directly follows the structure of Table 3 reported in Cooperdock et al. (2019) and Table 3 in Zeigler et al. (2023) to facilitate comparison with previous studies on apatite.**

**[b] Avg. 3D/2D is the average of all 3D/2D values in this study.**

**[c] Abs. avg. % diff. is the average absolute percent difference between the 2D and 3D data. We used the formula $\left(\frac{|2D-3D|}{2D}\right) \times 100$ to calculate the percent difference for consistency with Cooperdock et al. (2019) and Zeigler et al. (2023).**

## 5 Discussion

### 5.1 Accuracy and precision of 2D geometric data

The aim of this study was to use the approach of Zeigler et al. (2023) for apatite to develop corrections for systematic error and to assign uncertainties to geometric parameters estimated from microscopy measurements for the full range of zircon grains regularly dated by (U–Th)/He (Table 2). Previous studies of apatite reported the average 3D/2D value and its 1σ uncertainty as a measure of systematic error and reported the average absolute percent difference between the 2D and 3D data and its 1σ uncertainty as a measure of the uncertainty of each parameter (Cooperdock et al., 2019; Zeigler et al., 2023). For consistency with this past work, we additionally report our zircon results in this way (Table 3).





For V, tetragonal zircon 2D values overestimate the true grain volume (correction value of 0.81) with an
uncertainty of 13%, while ellipsoidal zircon 2D values underestimate the volume (correction value of
1.04) with an uncertainty of 21% (based on the regressions, Table 2). The average 3D/2D and average
absolute percent difference for our whole dataset are 0.88 and 19%, respectively (Table 3).

For $F_T$, our 2D values slightly overestimate, or are the same as, the 3D values. The isotope-specific
$^{238}F_T$ has a 2D correction of 0.97 for tetragonal grains and no correction for ellipsoidal grains, with
uncertainties of 2%-3% depending on geometry and size (Table 2). Our average 3D/2D value for $^{238}F_T$
is 0.98, with an absolute average difference of 3% (Table 3).

For $R_{FT}$, 2D measurements are systematically larger than 3D measurements (correction values of 0.92
and 0.98), with an uncertainty of 8% for both tetragonal and ellipsoidal grains (Table 2). Our average
3D/2D value for $R_{FT}$ is 0.94, with an average difference of 8% (Table 3).

We find that the same parameters control the corrections and uncertainties for zircon (this study) as for
apatite (Zeigler et al., 2023). Grain geometry is the primary control on the corrections for systematic
error. The choice of categorizing zircon as GEM A, B, or C dictates the choice of idealized geometry
(tetragonal or ellipsoidal), which in turn determines the correction. The correction for tetragonal grains
is larger than for ellipsoidal grains for all parameters, indicating that the tetragonal idealized geometry
does a systematically poorer job than the idealized ellipsoidal geometry at representing the true grain
morphology. Part of the reason for this is that the equations used for the idealized tetragonal geometry
assume a specific angle (45°), length, and shape for the terminations (Ketcham et al., 2011), while the
terminations on real zircon grains have a variety of shapes and angles.

The uncertainties are controlled primarily by grain geometry, while grain size is a second-order control
for the $F_T$ uncertainty only. The pattern of smaller uncertainties for tetragonal than ellipsoidal grains
(Table 2) implies that there is less variability in the morphology of tetragonal than ellipsoidal zircon
grains. This may be related to ellipsoidal zircon grains commonly being detrital and therefore more
likely to have irregularities than their pristine tetragonal counterparts. For $F_T$, grain size exerts an
additional influence on the uncertainty of tetragonal zircon, decreasing from 3% to 2% for medium and
small (maximum width <100µm) and large (maximum width >100µm) grains, respectively, with an
uncertainty of 3% for ellipsoidal grains of all sizes. The influence of grain size on $F_T$ uncertainty is
expected because the uncertainty on a microscopy measurement is proportionally larger for smaller
measurements.

Overall, the corrections and uncertainties for zircon (this work) are similar to or smaller than those for
apatite (Zeigler et al., 2023). We attribute this pattern to the greater ease of acquiring an accurate 2D
microscopy measurement of the zircon minimum width than the apatite minimum width. For tetragonal
zircon, the flat-sided habit makes it straightforward to roll the grain 90° into a stable position for the
minimum width measurement. However, for hexagonal apatite, the more rounded habit makes it
challenging to stabilize the grain for a minimum width measurement, resulting in greater uncertainty
and error in this 2D value and the 2D parameters computed from this measurement.





### 5.2 Implications: how much do the zircon geometric corrections uncertainties matter?

#### 5.2.1 Overview

To determine how much the geometric corrections and uncertainties (Table 2) affect the values and uncertainties in real ZHe data, we follow the approach of Zeigler et al. (2023) for apatite and apply our

corrections and uncertainties to the V, $F_T$, and $R_{FT}$ values of representative zircon grains from five samples (N = 28), four of which were used in this study and all of which were previously dated in the CU TRaIL (Tables D1–D3). This set of zircon includes tetragonal and ellipsoidal grains with a range of sizes. We then use the corrected V and isotope-specific $F_T$ values to calculate the mass, eU, and the corrected ZHe date, and propagate the geometric uncertainties in V and $F_T$ into the uncertainties of these

derived values. HeCalc (Martin et al., 2023) was used for uncertainty propagation into the corrected ZHe date assuming fully correlated (r = 1) isotope-specific $F_T$ uncertainties, which is the conservative approach that yields the maximum uncertainty. We then compare the geometric correction method (GCM) values and uncertainties in all parameters with their 2D uncorrected counterparts (Sect. 5.3.2–5.3.5) and generate corrected ZHe date vs. eU plots with both the GCM and 2D values (Fig. 8).


The average GCM/2D values for this dataset are in Table 4. All uncertainties in Table 4 and the discussion below are reported at 1σ. Over the last several years, standard practice in the CU TRaIL has been to report 15% 1σ uncertainties in eU based on estimates by Baughman et al. (2017). However, how eU uncertainties are reported varies for different labs, with no uncertainty commonly reported for eU

data. Therefore, no uncertainty is shown for $eU_{2D}$ values in the top plot for each sample in Fig. 8, and none is reported in Table D1.

#### 5.2.2 Mass and eU

eU is calculated from the parent isotope concentrations, which are computed using the absolute amounts

of the parent isotopes and the zircon mass. Mass is computed from V assuming a zircon density (here we use 4.65 g/cm$^3$). Conventionally the grain mass reported by labs has had no uncertainty attached to it because the geometric uncertainty in V (and thus on mass) was not well-known. By applying a correction factor to V based on grain geometry (0.81 or 1.04) and calculating mass using the corrected V, the mass$_{GCM}$ decreases for tetragonal grains and increases for ellipsoidal grains by the same

correction factor as the volume. The mass then inherits the same percent uncertainty as volume (13% or 21%, 1σ, depending on geometry).

For eU, the smaller mass$_{GCM}$ values for tetragonal grains (relative to mass$_{2D}$) mean larger $eU_{GCM}$ values (relative to $eU_{2D}$), while the larger mass$_{GCM}$ values for ellipsoidal grains (relative to mass$_{2D}$) mean

smaller $eU_{GCM}$ values (relative to $eU_{2D}$). In our example dataset (Table 4), the average $eU_{GCM}$ / $eU_{2D}$ is 1.23 for tetragonal grains and 0.96 for ellipsoidal grains. We propagated the analytical uncertainties in the parent isotopes only, as well as on both the parent isotopes and geometric uncertainties, into the $eU_{GCM}$ values. Including parent isotope uncertainties only yields average eU uncertainty values of 3% and 5% for tetragonal and ellipsoidal, respectively. Propagating both analytical and geometric





uncertainties yields average uncertainties of 12% and 20% for tetragonal and ellipsoidal zircon, respectively.

**Table 4. The average percent difference between the 2D and GCM values for example dataset of Tables D1-D3.**

| Parameter and Geometry[a] | Avg. GCM/2D[b] | % Total analytical uncertainty (TAU) only[c], 1σ | | | % TAU + Geometric (GMC) Uncertainty[d], 1σ | | | Avg. % uncert. increase[e], 1σ |
|---|---|---|---|---|---|---|---|---|
| | | Avg. | Min (%) | Max (%) | Avg. | Min (%) | Max (%) | |
| Mass | | | | | | | | |
| Tet. | 0.81 | NA | NA | NA | 13% | 13% | 13% | NA |
| Ellip. | 1.04 | NA | NA | NA | 21% | 21% | 21% | NA |
| eU | | | | | | | | |
| Tet. | 1.23 | 3% | 2% | 5% | 12% | 11% | 14% | 9% |
| Ellip. | 0.96 | 5% | 3% | 7% | 20% | 20% | 21% | 15% |
| Combined $F_T$ | | | | | | | | |
| Tet. | 0.97 | 2% | 1% | 8% | 3% | 2% | 8% | 1% |
| Ellip. | 1.00 | 4% | 1% | 6% | 5% | 3% | 7% | 1% |
| Corr. Date | | | | | | | | |
| Tet. | 1.03 | 3% | 2% | 5% | 4% | 3% | 5% | 1% |
| Ellip. | 1.00 | 4% | 3% | 6% | 5% | 4% | 7% | 1% |
| $R_{FT}$ | | | | | | | | |
| Tet. | 0.92 | NA | NA | NA | 8% | 8% | 8% | NA |
| Ellip. | 0.98 | NA | NA | NA | 8% | 8% | 8% | NA |


**n/a indicates "not applicable", for example, mass doesn't have any analytical uncertainty on the parent isotopes.**

**[a] There are N = 24 tetragonal and N = 4 ellipsoidal grains.**

**[b] The average of the GCM parameter (calculated using the GCM values) divided by the average of the 2D values (calculated using the 2D values) for the example data in Tables D1-D3. Values less than 1 indicate that the 2D value is larger than the GCM value. Values greater than 1 indicate that the 2D value is smaller than the GCM value.**


**[c] The average of the percent total analytical uncertainties (TAU) (i.e., parent isotope) only for the example data in Tables D1-D3.**

**[d] The average of the percent TAU + geometric (GCM) uncertainties for the example data in Tables D1-D3.**

**[e] The average percent increase is the difference between the TAU only and TAU + GCM uncertainties.**

### 5.2.3 Combined $F_T$ values

The combined $F_T$ values are calculated using both the parent isotope amounts and the isotope-specific $F_T$ values. For our example dataset, we apply the correction factors in Table 2 based on grain geometry and size to the isotope-specific $F_T$ values and then use these corrected values to compute the combined $F_{T,GCM}$ value. $F_{T,GCM}$ is smaller than $F_{T,2D}$ for tetragonal grains and is the same as $F_{T,2D}$ for ellipsoidal grains ($F_{T,GCM}/F_{T,2D}$ = 0.97 and 1 for tetragonal and ellipsoidal grains; Table 4).


$F_T$ values have not typically been reported with an uncertainty because the geometric uncertainty on $F_T$ has been poorly constrained until now. We propagated uncertainties into the combined $F_T$ value using the parent isotope uncertainties only, as well as using both parent isotope and geometric uncertainties. For the example dataset, including analytical uncertainties only yields average uncertainties in the





combined $F_T$ value of 2% and 4% for tetragonal and ellipsoidal zircon, respectively. Propagating both parent isotope and geometric uncertainties generates average values of 3% and 5% for the two geometries.

### 5.2.4 Corrected zircon (U-Th)/He dates

We calculated $F_T$-corrected ZHe dates iteratively with an age equation that incorporates the isotope-specific $F_T$ corrections (Ketcham et al., 2011). For the ZHe dates of our example dataset, the smaller $F_{T,GCM}$ values for tetragonal grains (relative to $F_{T,2D}$) mean larger corrections for alpha ejection and date$_{GCM}$ values that are older than the date$_{2D}$ values (avg. date$_{GCM}$/date$_{2D}$ = 1.03), while for ellipsoidal grains the ZHe date$_{GCM}$ values are unchanged from the date$_{2D}$ values (avg. date$_{GCM}$/date$_{2D}$ = 1.00).


We calculated the uncertainty in the corrected (U–Th)/He dates first by propagating the analytical uncertainties in the parent and daughter only and then by additionally including the geometric uncertainties in the isotope-specific $F_{T,GCM}$ values and assuming fully correlated $F_{T,GCM}$ uncertainties (Table 4). For this dataset, propagating only analytical uncertainties yields average uncertainties of 3%
and 4% for tetragonal and ellipsoidal grains, respectively. Including both analytical and geometric uncertainties yields average uncertainties of 4% and 5% for the two geometries.

### 5.2.5 $R_{FT}$

We applied the correction factors based on grain geometry in Table 2 to $R_{FT}$ values from the example dataset. The $R_{FT,GCM}$ values are always smaller than $R_{FT,2D}$ values ($R_{FT,GCM}$/$R_{FT,2D}$ = 0.92 and 0.98 for
tetragonal and ellipsoidal grains) (Table 4). The uncertainty in $R_{FT}$ is 8% (1σ) for both geometries. $R_{FT}$ is used during thermal history modeling and this uncertainty should be included in modeling when possible.

### 5.2.6 Summary

Like for apatite, correcting ZHe data for systematic error and propagating appropriate geometric
uncertainties has substantial influence on eU but less influence on the ZHe date. For eU, the GCM values of the example dataset increase by 9%–15%, causing a noticeable shift of data to the right (higher eU values) on the date–eU plots (compare the top and bottom plots for each sample in Fig. 8). When both analytical and geometric uncertainties are included, eU uncertainties average 12% and 20% for the different grain geometries, indicating the importance of appropriately reporting,
representing, and considering eU uncertainties when interpreting ZHe datasets. For the corrected ZHe date, for tetragonal grains the ZHe date$_{GCM}$ values average 3% older than the date2D values, with no change for ellipsoidal grains. Typical ZHe date uncertainties increase by only 1% for both geometries when geometric uncertainties are propagated in addition to analytical uncertainties.





**Figure 8. ZHe date-eU plots for five samples previously dated in the CU TRaIL showing the effects of corrections and uncertainty estimates on typical ZHe data. The top plot for each sample ("2D") are ZHe date$_{2D}$ vs. eU$_{2D}$ plots with only analytical uncertainties propagated into the date uncertainty and no eU uncertainty shown. The bottom plot for each sample ("GCM") are ZHe date$_{GCM}$ vs. eU$_{GCM}$ plots with both analytical and geometric uncertainties propagated into the date uncertainty and geometric uncertainties included on eU uncertainty. When uncertainty bars are not visible, they are on the order of the symbol size. An idealized tetragonal geometry was used for 2D geometric parameter calculations for the zircon represented by purple circles, while an idealized ellipsoidal geometry was used for the zircon represented by green circles.**



### 5.3 The Zircon Geometric Correction Method: a practical workflow

Like for apatite (Zeigler et al., 2023), the Geometric Correction Method for zircon shown in Fig. 9 can be easily integrated into existing (U–Th)/He dating workflows with no additional time, cost, or

equipment. The final corrections and uncertainties are most appropriate for grains with characteristics like those used in this calibration study, with microscopy measurements and 2D calculations done as in this work. Zircon grains should have geometries like those in Figure 3, length/maximum width ratios of 1.0-8.0, minimum width/maximum width ratios of 1.0-1.9, and maximum widths between 34 μm and 160 μm. All equations required for the calculations are in Appendix A. The corrections for systematic

error and uncertainties reported here are only those from grain geometry. For $F_T$, additional inaccuracy and uncertainty may be caused by parent isotope zonation (e.g., Farley et al., 1996, Hourigan et al., 2005), grain abrasion (e.g., Rahl et al., 2003), and grain breakage (He and Reiners, 2022), which have potential to be accounted for separately. For mass and the derived eU values, additional uncertainty may be introduced by radiation damage, which can cause the zircon density (used to calculate mass) to drop

by up to 16% (e.g., Holland and Gottfried, 1955). The following workflow is the same as that for apatite, but modified slightly for zircon.

Step 1. Select zircon grain geometry and GEM category.

Choose a zircon grain for analysis. Decide whether the grain is tetragonal or ellipsoidal, which is all that is required to correct the 2D values and assign uncertainty. However, we encourage taking additional notes on the zircon clarity and other characteristics (Fig. 3, Fig. B1), which can be helpful for data interpretation.

Step 2. Measure the zircon.

Measure the zircon using the procedure outlined in Sect. 3.2 and Fig. 4.
- Measure the grain length parallel to the c-axis. Only a single length is required, but if the grain has an extremely angled or uneven end then measuring and averaging two lengths may better

545       capture the average length.
- Measure the zircon grain's maximum width, which is perpendicular to the grain length. Note that the maximum width is a factor for selecting the proper $F_{T,GCM}$ uncertainty (see Step 5; Table 2).
- Rotate the zircon 90° and measure the zircon's minimum width.


Step 3. Calculate the zircon's 2D values.

Calculate 2D microscopy V and isotope-specific $F_T$ values using the tetragonal or ellipsoidal equations of Ketcham et al. (2011) and calculate $R_{FT}$ using the equations of Cooperdock et al. (2019). The parent

isotope data must first be acquired for the $F_T$ and $R_{FT}$ values to be computed.

Step 4. Correct the 2D values.





Multiply the 2D microscopy V, isotope-specific $F_T$, and $R_{FT}$ values by the correction factor based on the grain geometry to produce the $V_{GCM}$, $F_{T,GCM}$, and $R_{FT,GCM}$ values (Table 2).

Step 5. Assign uncertainty.

Assign the uncertainty value to each parameter according to the grain geometry (for $V_{GCM}$, $F_{T,GCM}$, $R_{FT,GCM}$) and maximum width (for $F_{T,GCM}$) (Table 2).

Step 6. Calculate derived parameters and propagate uncertainties.

• Compute mass and eU using the $V_{GCM}$ values. Uncertainty in V should be propagated into the uncertainties in these derived parameters.
• Compute corrected (U–Th)/He dates using the isotope-specific $F_{T,GCM}$ values. Uncertainty in $F_T$ should be propagated into the final uncertainty in the corrected He date. This uncertainty propagation can be accomplished, for example, by using the open-access Python program HeCalc for (U–Th)/He data reduction (Martin et al., 2023).

For example: a zircon selected for analysis has a maximum width of 89 μm, a GEM value of A, and a $^{238}F_{T,2D}$ value of 0.81 (see Appendix A and the footnotes of Tables D1–D3 for the details of this calculation). The analyst uses Table 2 to select the correction for tetragonal grains (0.97) and calculates $F_{T,GCM}$ = $F_{T,2D}$ × correction = 0.81 × 0.97 = 0.78. The analyst then selects the proper uncertainty from
Table 2: this tetragonal grain is considered medium-sized because it is 89 μm wide, so it has a geometric uncertainty of 3%. The final $^{238}F_{T,GCM}$ = 0.78 ± 3 % if only the geometric uncertainty is propagated into the $^{238}F_T$ value. This procedure is repeated for each isotope-specific $F_{T,2D}$ value. The isotope-specific $F_{T,GCM}$ values are used to calculate the corrected ZHe date, and both the uncertainty in each isotope-specific $F_T$ and the analytical uncertainty in the parent and daughter isotopes are
propagated into the uncertainty in the corrected (U–Th)/He date.

**6 Conclusions**

In this paper we develop a set of corrections for systematic error and assign uncertainties to zircon geometric parameters calculated from 2D microscopy measurements. The uncertainties in these geometric parameters (V, isotope-specific $F_T$ values) and the data derived from them (mass, eU,
combined $F_T$, corrected (U-Th)/He date, $R_{FT}$) have not traditionally been included in reported uncertainties in ZHe data, but are important for appropriate representation and interpretation of such datasets. This study builds on the work of Zeigler et al. (2023) for apatite, and similarly presents the only no-cost, easy-to implement, and backwards-compatible solution to this problem, but for zircon. It is straightforward to incorporate the Geometric Correction Method (GCM) into existing workflows
(Fig. 9) and to apply it to previously published data. These corrections and uncertainties are most appropriate for zircon grains like those in this calibration study, with microscopy measurements and



parameter calculations performed as in this work. The corrections and uncertainties in this study were derived from the regression of 2D and 3D measurements of 223 zircon grains displaying the range of morphologies commonly dated by (U–Th)/He. The derived corrections and uncertainties were then

applied to real ZHe data to determine their typical impact. The key outcomes are:

1. Both uncertainty and systematic error are associated with the microscopy approach to calculating V, $F_T$, and $R_{FT}$ for zircon, but the magnitudes are slightly smaller than they are for apatite.

2. Using 2D microscopy measurements, the true values of V are overestimated for tetragonal grains and underestimated for ellipsoidal grains; the true values of $F_T$ are slightly overestimated (tetragonal zircon) or correctly determined (ellipsoidal zircon); and the true values of $R_{FT}$ are overestimated for both geometries.

3. All corrections for systematic error are larger for tetragonal than ellipsoidal grains, but all

uncertainties are the same or smaller for tetragonal than ellipsoidal grains. V has the largest magnitude of overestimation and uncertainty, followed by $R_{FT}$ and then $F_T$.

4. For a subset of real ZHe data (N = 28 analyses), the correction factor for eU typically increases the eU by ~20% (for tetragonal grains) and decreases eU by ~4% (for ellipsoidal grains) with associated 1σ uncertainties of 12%-20% when both analytical and geometric uncertainties are

included. These shifts in eU values and the uncertainty magnitudes are substantial and should be considered when interpreting ZHe data.

5. For the real dataset, the correction factor for the corrected (U–Th)/He date generally increases the date by 3% for tetragonal grains with associated 1σ uncertainties of 4%–5% if both analytical and geometric uncertainties are included. Application of the GCM to ellipsoidal grains

does not change the corrected ZHe date but does increase the associated 1σ uncertainties by 1%.

6. The geometric corrections and geometric uncertainties are substantial enough, while being simple enough to account for, that they should be routinely included when reporting eU and corrected ZHe dates.





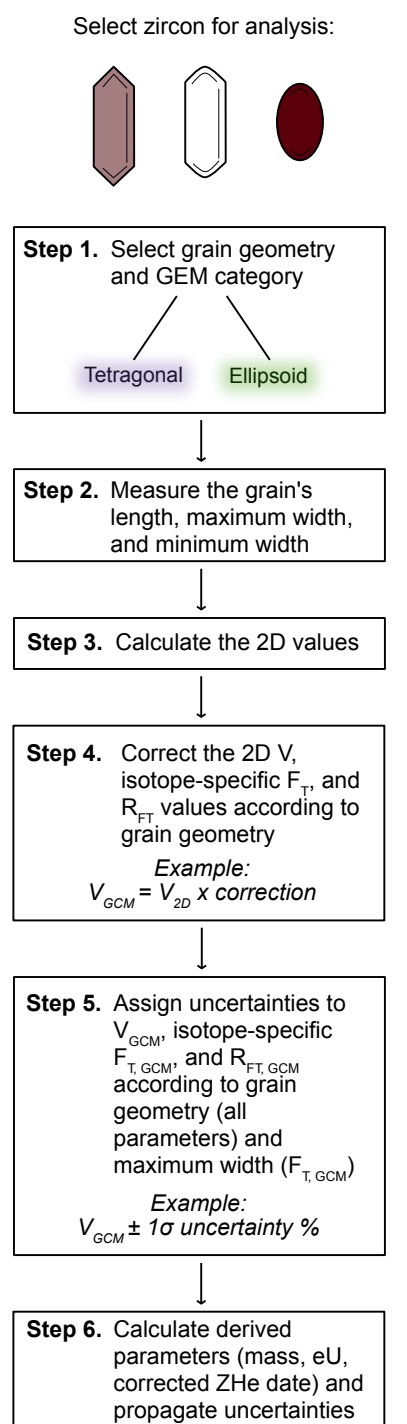


**Figure 9. Flow chart outlining workflow for the Geometric Correction Method.**



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

## Appendix A

All equations necessary to use the corrections and uncertainties are listed below.

Equations A1 to A4 are for a tetragonal (GEM A or B) grain from Ketcham et al. (2011), where we use
$W_{min}$ instead of a, $W_{max}$ instead of b, and L instead of c. Here, S is the weighted mean stopping distance of an alpha particle for a given parent isotope decay chain (15.55, 18.05, 18.43, and 4.76 μm for $^{238}$U, $^{235}$U, $^{232}$Th, and $^{147}$Sm, respectively), $R_{SV}$ is the SV-equivalent spherical radius, and Np is the number of pyramidal terminations. Equation A4 is used to calculate each isotope-specific $F_T$ value, each with a different stopping distance (S).

$$V = W_{min}W_{max}L - N_p \frac{W_{min}}{4}\left(W_{max}^2 + \frac{W_{min}^2}{3}\right) \text{ where } W_{min} \leq W_{max} \qquad \text{(A1)}$$

$$SA = 2(W_{min}W_{max} + W_{max}L + W_{min}L) - N_p\left(\frac{W_{min}^2 + W_{max}^2}{2} + \left(2 - \sqrt{2}\right)W_{min}W_{max}\right) \qquad \text{(A2)}$$





$$R_{SV} = \frac{3V}{SA} \tag{A3}$$

$$F_T = 1 - \frac{3}{4}\frac{S}{R_{SV}} + \left(0.2095(W_{min} + W_{max} + L) - \left(0.096 - 0.013\frac{W_{min}^2 + W_{max}^2}{L^2}\right)(W_{min} + W_{max})N_p\right)\frac{S^2}{V}$$
(A4)

Equations A5 to A8 are for an ellipsoidal grain (GEM C) from Ketcham et al. (2011). Equation A8 is used to calculate each isotope-specific $F_T$ value, each with a different stopping distance.

$$V = \frac{4}{3}\pi W_{min}W_{max}L \tag{A5}$$

$$SA = 4\pi\left(\frac{W_{min}^p W_{max}^p + W_{max}^p L^p + L^p W_{min}^p}{3}\right)^{1/p} \text{ with } p = 1.6075 \tag{A6}$$

$$R_{SV} = \frac{3V}{SA} \tag{A7}$$

$$F_T = 1 - \frac{3}{4}\frac{S}{R_{SV}} + \left[\frac{1}{16} + 0.1686\left(1 - \frac{W_{min}}{R_{SV}}\right)^2\right]\left(\frac{S}{R_{SV}}\right)^3 \tag{A8}$$

The age equation from Ketcham et al. (2011) is as follows.

$$^4He = 8F_{T,238}\,^{238}U(e^{\lambda_{238}t} - 1) + 7F_{T,235}\,^{235}U(e^{\lambda_{235}t} - 1)$$

$$+ 6F_{T,232}\,^{232}Th(e^{\lambda_{232}t} - 1) + F_{T,147}\,^{147}Sm(e^{\lambda_{147}t} - 1) \tag{A9}$$

Equations A10 to A15 are for combined $F_T$ and $R_{FT}$ from Cooperdock et al. (2019). Here, $S_{238}$, $S_{232}$, and $S_{235}$ are the weighted mean stopping distances for each decay chain in zircon, using the values noted above. $A_{238}$ and $A_{232}$ are the activities of $^{238}U$ and $^{232}Th$, respectively.

$$\frac{S}{R} = 1.681 - 2.428\overline{F_T} + 1.153\overline{F_T^2} - 0.406\overline{F_T^3} \tag{A10}$$

$$A_{238} = (1.04 + 0.247[Th/U])^{-1} \tag{A11}$$

$$A_{232} = (1 + 4.21/[Th/U])^{-1} \tag{A12}$$

$$\overline{F_T} = A_{238}F_{T,238} + A_{232}F_{T,232} + (1 - A_{238} - A_{232})F_{T,235} \tag{A13}$$

$$\overline{S} = A_{238}S_{238} + A_{232}S_{232} + (1 - A_{238} - A_{232})S_{235}, \tag{A14}$$



$$R_{FT} = \overline{S} / \left(\frac{S}{R}\right) \tag{A15}$$

Equation A16 is for eU from Cooperdock et al. (2019).

$\quad eU = [U] + 0.238[Th] + 0.0012[Sm] \ (or \ 0.0083[^{147}Sm]) \tag{A16}$



## Appendix B: Additional sample and method information

(a)

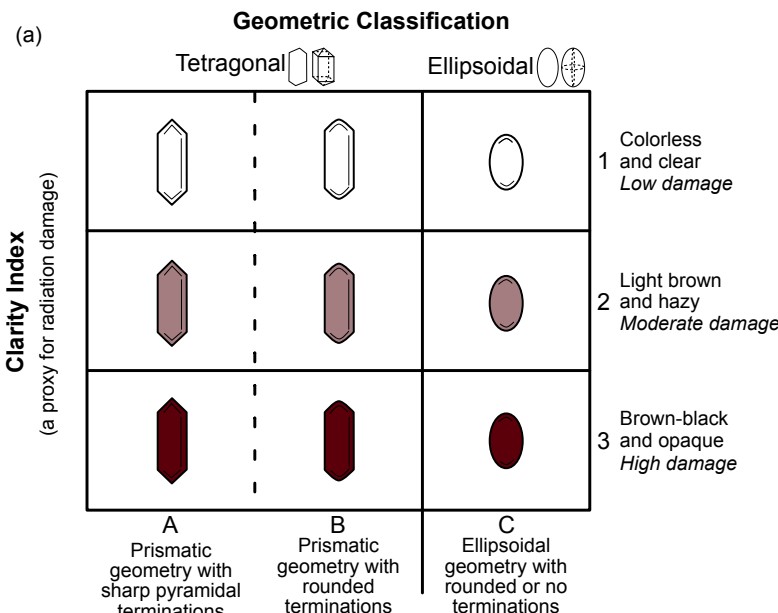

(b)

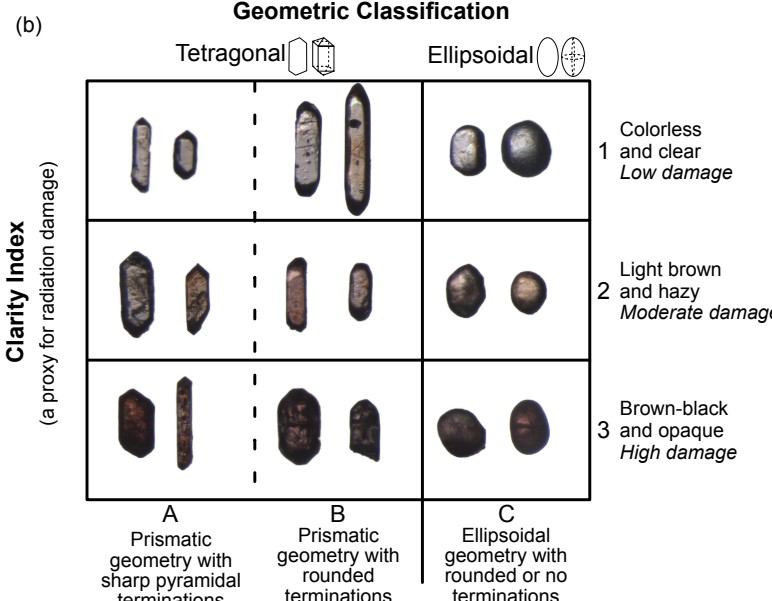

**Figure B1. The expanded Grain Evaluation Matrix (GEM) for zircon in (a) schematic form and (b) with images of real zircon analyzed in this study. The geometric classification axis is the same as in Figure 3. The clarity index axis measures the zircon color and opacity as a qualitative proxy for radiation damage. Darkness and opacity increase from 1 (colorless, clear), 2 (light brown and hazy) to 3 (brown-black, opaque). The GEM in Figure 3 collapses the clarity axis since radiation damage does not influence the regressions. Users are encouraged to note the color and clarity of the zircon grain as a qualitative proxy for radiation damage, which bears on the interpretation of ZHe data. Grains can be described by combining a geometric value and a clarity value (e.g., A1, B2).**



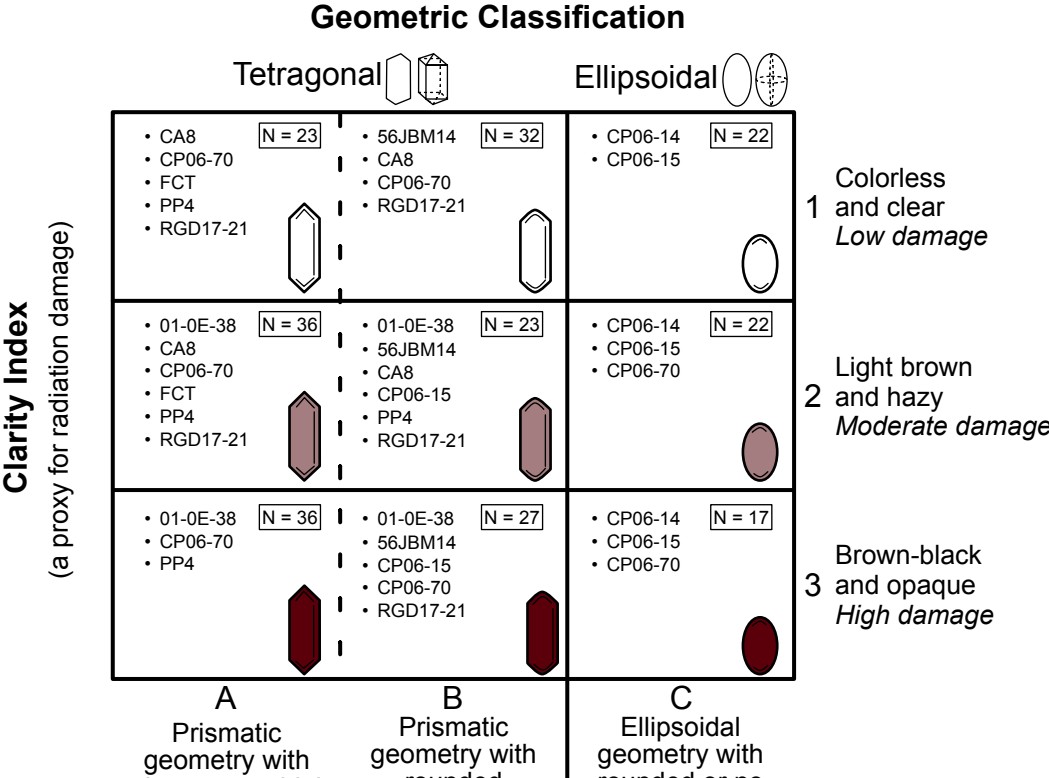

**Figure B2.** The expanded Grain Evaluation Matrix listing the samples and number of grains for which high-quality CT data (N = 223) were acquired in each category in this study.






**Table B1. Zircon CT scan parameters**

| Mount[a] | 1 | 3 | 4 | 5 | 6 | 7 |
|---|---|---|---|---|---|---|
| Objective | 4X | 4X | 4X | 4X | 4X | 4X |
| Pixel Size (µm) | 0.88 | 0.92 | 0.92 | 0.84 | 0.84 | 0.84 |
| X-Ray Power (W) | 3 | 10 | 10 | 10 | 10 | 10 |
| X-Ray Voltage (kV) | 40 | 120 | 120 | 140 | 140 | 140 |
| Number of Projections | 3201 | 2401 | 2401 | 2401 | 2401 | 2401 |
| Binning | 1 | 1 | 1 | 1 | 1 | 1 |
| Filter[b] | Air | HE1 | HE1 | HE1 | HE1 | HE1 |
| Height (pixels) | 2026 | 2026 | 2026 | 2026 | 2026 | 2026 |
| Width (pixels) | 2026 | 2026 | 2026 | 2026 | 2026 | 2026 |
| Sample Theta (°) | -180 | -180 | -180 | -180 | -180 | -180 |
| Detector To Sample Distance (mm) | 17.0 | 40.0 | 40.0 | 25.5 | 25.5 | 25.5 |
| Source To Sample Distance (mm) | -6.0 | -15.0 | -15.0 | -8.5 | -8.5 | -8.5 |
| Exposure (s) | 3 | 1 | 1 | 1 | 1 | 1 |
| Total Scan Time (h) | 4.8 | 2.3 | 2.3 | 2.3 | 2.3 | 2.3 |


[a] **Mount 2 is excluded from this dataset (see Sect. 3.3 for details).**

[b] **HE1 is a filter provided by Zeiss that is used to avoid beam hardening artefacts.**





## Appendix C: Additional regression and uncertainty information







**Figure C1. Plots illustrating how the corrections for systematic error (a-c) and how uncertainties (d-f) were determined for different parent isotope-specific $F_T$ values. This figure is the same as Figure 7b and 7e for $^{238}F_T$, but for the $^{235}F_T$, $^{232}F_T$, and $^{147}F_T$ values.**

**Table C1. Results of Tukey's Highly Significant Difference[a] test to determine if different groups of grains have statistically different slopes.**

| Grouping & Pairs | Difference in Slopes | 95% CI[b] | Adjusted p-value[c] |
|---|---|---|---|
| **Volume** | | | |
| GEM: Geometric Classification | | | |
| B-A | < 0.001 | [-0.001, 0.001] | 1 |
| **C-A** | **-0.276** | **[-0.276, -0.276]** | **< 0.001** |
| **C-B** | **-0.276** | **[0.153, 0.153]** | **< 0.001** |
| Size | | | |
| Small&Medium-Large | 0.004 | [-0.041, 0.032] | 0.818 |
| GEM: Clarity Index | | | |
| 1-2 | 0.004 | [-0.043, 0.050] | 0.976 |
| 1-3 | 0.007 | [-0.043, 0.056] | 0.945 |
| 2-3 | 0.003 | [-0.046, 0.051] | 0.99 |
| $^{238}F_T$ | | | |
| GEM: Geometric Classification | | | |
| B-A | < 0.001 | [-0.001, 0.001] | 0.922 |
| **C-A** | **-0.025** | **[-0.025, -0.025]** | **< 0.001** |
| **C-B** | **-0.025** | **[-0.025, -0.025]** | **< 0.001** |
| Size | | | |
| Small&Medium-Large | < 0.001 | [-0.004, 0.005] | 0.971 |
| GEM: Clarity Index | | | |
| 1-2 | < 0.001 | [-0.004, 0.005] | 0.979 |
| 1-3 | < 0.001 | [-0.004, 0.005] | 0.945 |
| 2-3 | < 0.001 | [-0.004, 0.005] | 0.99 |
| $R_{FT}$ | | | |
| GEM: Geometric Classification | | | |
| B-A | 0 | [-0.001, 0.001] | 1 |
| **C-A** | **0.063** | **[0.063, 0.063]** | **< 0.001** |
| **C-B** | **0.063** | **[0.063, 0.063]** | **< 0.001** |
| Size | | | |
| Small&Medium-Large | < 0.001 | [-0.009, 0.007] | 0.818 |
| GEM: Clarity Index | | | |
| 1-2 | < 0.001 | [-0.01, 0.01] | 0.979 |
| 1-3 | < 0.001 | [-0.01, 0.01] | 0.945 |
| 2-3 | < 0.001 | [-0.01, 0.01] | 0.99 |

**[a] Tukey's Highly Significant Difference tests if slopes are significantly different from each other or not and takes into account the uncertainties on the slopes. Where the null hypothesis, $H_0$, is $\beta_1 = \beta_2$ and the alternative hypothesis, $H_1$, is $\beta_1 \neq \beta_2$.**

**[b] The 95% confidence interval (CI) of the difference in slopes.**

**[c] A p-value < 0.05 indicates that $H_0$ can be rejected, i.e., there is a significant difference between the slopes of the pair. If the p-value is > 0.05, this indicates that there is no significant difference between the means of the pair. Bolded pairs of slopes are those with p-values <0.05 and therefore are treated as separate groups.**





**Table C2. Uncertainty values (1σ) for different groupings of physical variables.**

| Geometry | Size[a] | Clarity | N | Uncertainty |
|---|---|---|---|---|
| **Volume** | | | | |
| **Tet.** | **All sizes** | **1 & 2 & 3** | **162** | **13%** |
| Tet. | Small & Medium | 1 & 2 & 3 | 117 | 13% |
| Tet. | Small & Medium | 1 | 46 | 12% |
| Tet. | Small & Medium | 2 | 39 | 14% |
| Tet. | Small & Medium | 3 | 32 | 12% |
| Tet. | Large | 1 & 2 & 3 | 45 | 13% |
| Tet. | Large | 1 | 9 | 14% |
| Tet. | Large | 2 | 20 | 12% |
| Tet. | Large | 3 | 16 | 12% |
| **Ellip.** | **All sizes** | **1 & 2 & 3** | **61** | **21%** |
| Ellip. | Small & Medium | 1 & 2 & 3 | 45 | 21% |
| Ellip. | Small & Medium | 1 | 18 | 22% |
| Ellip. | Small & Medium | 2 | 15 | 11% |
| Ellip. | Small & Medium | 3 | 12 | 21% |
| Ellip. | Large | 1 & 2 & 3 | 16 | 13% |
| Ellip. | Large | 1 | 4 | 12% |
| Ellip. | Large | 2 | 7 | 13% |
| Ellip. | Large | 3 | 5 | 3% |
| **$^{238}F_T$** | | | | |
| Tet. | All sizes | 1 & 2 & 3 | 162 | 3% |
| **Tet.** | **Small & Medium** | **1 & 2 & 3** | **117** | **3%** |
| Tet. | Small & Medium | 1 | 46 | 3% |
| Tet. | Small & Medium | 2 | 39 | 3% |
| Tet. | Small & Medium | 3 | 32 | 3% |
| **Tet.** | **Large** | **1 & 2 & 3** | **45** | **2%** |
| Tet. | Large | 1 | 9 | 1% |
| Tet. | Large | 2 | 20 | 2% |
| Tet. | Large | 3 | 16 | 2% |
| **Ellip.** | **All sizes** | **1 & 2 & 3** | **61** | **3%** |
| Ellip. | Small & Medium | 1 & 2 & 3 | 45 | 3% |
| Ellip. | Small & Medium | 1 | 18 | 3% |
| Ellip. | Small & Medium | 2 | 15 | 3% |
| Ellip. | Small & Medium | 3 | 12 | 3% |
| Ellip. | Large | 1 & 2 & 3 | 16 | 1% |
| Ellip. | Large | 1 | 4 | 2% |
| Ellip. | Large | 2 | 7 | 1% |
| Ellip. | Large | 3 | 5 | 1% |
| **$R_{FT}$** | | | | |
| **Tet.** | **All sizes** | **1 & 2 & 3** | **162** | **8%** |
| Tet. | Small & Medium | 1 & 2 & 3 | 117 | 7% |
| Tet. | Small & Medium | 1 | 46 | 7% |
| Tet. | Small & Medium | 2 | 39 | 8% |
| Tet. | Small & Medium | 3 | 32 | 7% |
| Tet. | Large | 1 & 2 & 3 | 45 | 8% |
| Tet. | Large | 1 | 9 | 4% |
| Tet. | Large | 2 | 20 | 7% |
| Tet. | Large | 3 | 16 | 8% |
| **Ellip.** | **All sizes** | **1 & 2 & 3** | **61** | **8%** |
| Ellip. | Small & Medium | 1 & 2 & 3 | 45 | 8% |
| Ellip. | Small & Medium | 1 | 18 | 9% |
| Ellip. | Small & Medium | 2 | 15 | 8% |
| Ellip. | Small & Medium | 3 | 12 | 8% |
| Ellip. | Large | 1 & 2 & 3 | 16 | 5% |
| Ellip. | Large | 1 | 4 | 8% |
| Ellip. | Large | 2 | 7 | 4% |
| Ellip. | Large | 3 | 5 | 3% |


[a] **Groups in bold are the groups for which uncertainties are reported (i.e., geometry only for V and $R_{FT}$; geometry and grain size for $F_T$).**



**Appendix D: Application of geometric parameter corrections and uncertainties to a real dataset**

**Table D1. Results of applying geometric corrections and uncertainties (1σ) to zircon (U-Th)/He data from a suite of samples previously dated in the CU TRaIL for mass and eU.**

| Sample and aliquot[a] | Geo.[b] | Max. Width[c] (µm) | Mass$_{2D}$[d] (µg) | Mass$_{GCM}$[e] (µg) | ±[f] (µg) | ±[g] (%) | eU$_{2D}$[h] (ppm) | eU$_{GCM}$[i] (ppm) | ±TAU[j] (ppm) | ±TAU[k] (%) | ±TAU + geom[l] (ppm) | ±TAU + geom[m] (%) |
|---|---|---|---|---|---|---|---|---|---|---|---|---|
| **RGD17-21** | | | | | | | | | | | | |
| z02 | Tet. | 119.5 | 26.8 | 21.7 | 2.8 | 13% | 805.1 | 987.3 | 32.2 | 3% | 123.7 | 13% |
| z03 | Tet. | 118.6 | 20.0 | 16.2 | 2.1 | 13% | 435.7 | 534.3 | 10.8 | 2% | 65.7 | 12% |
| **PP4** | | | | | | | | | | | | |
| z01 | Tet. | 90.5 | 7.6 | 6.2 | 0.8 | 13% | 61.3 | 75.5 | 2.8 | 4% | 8.8 | 12% |
| z02 | Tet. | 108.5 | 16.6 | 13.4 | 1.7 | 13% | 538.6 | 660.6 | 13.6 | 2% | 78.2 | 12% |
| z03 | Tet. | 121.4 | 16.4 | 13.2 | 1.7 | 13% | 943.8 | 1158.4 | 23.8 | 2% | 135.1 | 12% |
| **CA8** | | | | | | | | | | | | |
| z01 | Tet. | 118.3 | 11.3 | 9.1 | 1.2 | 13% | 87 | 107.2 | 3.0 | 3% | 13.2 | 12% |
| z02 | Tet. | 138.2 | 14.0 | 11.4 | 1.5 | 13% | 189.4 | 232.4 | 4.3 | 2% | 27.4 | 12% |
| z03 | Tet. | 140.3 | 8.5 | 6.9 | 0.9 | 13% | 219.1 | 271.2 | 7.5 | 3% | 32.2 | 12% |
| z04 | Ellip. | 85.1 | 2.7 | 2.8 | 0.6 | 21% | 1417 | 1353.7 | 42.0 | 3% | 265.0 | 20% |
| **BP19-14** | | | | | | | | | | | | |
| z01 | Tet. | 116.0 | 11.7 | 9.5 | 1.2 | 13% | 463.4 | 568.2 | 17.2 | 3% | 72.6 | 13% |
| z02 | Tet. | 78.0 | 3.7 | 3.0 | 0.4 | 13% | 456.9 | 560.4 | 26.2 | 5% | 74.0 | 13% |
| z03 | Ellip. | 140.0 | 8.7 | 9.1 | 1.9 | 21% | 1051 | 1003.2 | 27.7 | 3% | 201.0 | 20% |
| z04 | Tet. | 50.0 | 1.7 | 1.4 | 0.2 | 13% | 695.3 | 852.7 | 39.8 | 5% | 115.8 | 14% |
| z05 | Tet. | 104.0 | 16.1 | 13.0 | 1.7 | 13% | 616.3 | 755.5 | 31.5 | 4% | 99.4 | 13% |
| z06 | Ellip. | 139.0 | 24.7 | 25.7 | 5.4 | 21% | 1125 | 1074.3 | 71.0 | 7% | 225.8 | 21% |
| z07 | Tet. | 141.0 | 19.2 | 15.5 | 2.0 | 13% | 553.1 | 680.1 | 20.3 | 3% | 82.6 | 12% |
| z08 | Ellip. | 139.0 | 11.3 | 11.8 | 2.5 | 21% | 110.8 | 105.8 | 6.1 | 6% | 21.2 | 20% |
| z09 | Tet. | 41.0 | 0.9 | 0.7 | 0.1 | 13% | 202.1 | 247.7 | 9.6 | 4% | 32.7 | 13% |
| z10 | Tet. | 81.0 | 2.8 | 2.3 | 0.3 | 13% | 141.8 | 173.9 | 3.1 | 2% | 21.4 | 12% |
| z11 | Tet. | 149.0 | 8.7 | 6.9 | 0.9 | 13% | 423.2 | 528.1 | 17.4 | 3% | 67.9 | 13% |
| **FCT** | | | | | | | | | | | | |
| z36 | Tet. | 104.3 | 11.3 | 9.2 | 1.2 | 13% | 386.6 | 474.1 | 11.6 | 2% | 56.0 | 12% |
| z37 | Tet. | 69.7 | 3.5 | 2.8 | 0.4 | 13% | 460.7 | 565.1 | 12.9 | 2% | 65.9 | 12% |
| z38 | Tet. | 134.7 | 12.0 | 9.7 | 1.3 | 13% | 275.9 | 338.7 | 6.3 | 2% | 40.2 | 12% |
| z39 | Tet. | 63.8 | 2.8 | 2.2 | 0.3 | 13% | 516.4 | 633.8 | 12.9 | 2% | 74.4 | 12% |
| z40 | Tet. | 86.7 | 8.1 | 6.6 | 0.9 | 13% | 517.7 | 634.9 | 11.7 | 2% | 71.0 | 11% |
| z41 | Tet. | 110.1 | 9.1 | 7.4 | 1.0 | 13% | 387.4 | 475.5 | 8.7 | 2% | 55.1 | 12% |
| z42 | Tet. | 132.6 | 11.5 | 9.3 | 1.2 | 13% | 1842 | 2259.1 | 43.9 | 2% | 260.4 | 12% |
| z43 | Tet. | 108.4 | 9.7 | 7.9 | 1.0 | 13% | 229.2 | 281.3 | 6.5 | 2% | 33.2 | 12% |

All uncertainties reported at the 1σ level.

[a] All RGD17-21, PP4, CA8, and BP19-14 data are published in McGrew & Metcalf (2020), Havranek and Flowers (2022), Basler et al. (2021) and Peak et al., 2023, respectively.

[b] Geometry is defined as described in Figure 3 of Ketcham et al. (2011). All GEM A and B grains are tetragonal (tet.) ("tetrahedral prism" of Ketcham et al. (2011)) and all GEM C grains are ellipsoidal (ellip.).

[c] Maximum width is measured perpendicular to the length/c-axis.

[d] Mass$_{2D}$ is the mass of the crystal determined by 2D microscopy measurements, the volume assuming the reported grain geometry, and the volume equations and mineral densities in Ketcham et al. (2011).

[e] Mass$_{GCM}$ is computed the same as mass$_{2D}$, but the 2D V is corrected by applying the correction factor in Table 2 based on the grain geometry, and this new volume is used in the mass calculation.

[f] The 1σ uncertainty on mass$_{GCM}$ is calculated by propagating the uncertainty on V from Table 2 based on grain geometry through the mass equation.

[g] The 1σ percent uncertainty on mass$_{GCM}$.



**h eU$_{2D}$ is effective Uranium concentration calculated using the mass$_{2D}$. Calculated as U + 0.238\*Th + 0.0012\*Sm after equation A7 of Cooperdock et al. (2019).**

**i eU$_{GCM}$ is computed the same as eU$_{2D}$ but uses the mass$_{GCM}$ value.**

**j The 1σ total analytical uncertainty (TAU, which are the uncertainties on the parent isotopes) on eU. This calculation ignores the negligible contribution from Sm concentration uncertainty and uses 0% geometric uncertainty.**

**k The 1σ total analytical percent uncertainty on eU$_{GCM}$.**

**l The 1σ TAU + geometric uncertainty on eU$_{GCM}$. This uncertainty includes the total analytical uncertainty and the uncertainty assigned based on grain geometry (Table 2), assumes that the geometric uncertainties on U and Th concentrations are perfectly correlated (r = 1), and ignores the negligible contribution from Sm concentration uncertainty. Although the correlation coefficient**
**will vary with each data set, the dominant contribution to concentration uncertainty comes from the volumetric uncertainty, which is highly correlated. Additionally, assuming perfect correlation yields the maximum possible value, so we use this conservative approach.**

**m The 1σ total analytical + geometric percent uncertainty on eU$_{GCM}$.**

**Table D2. Results of applying geometric corrections and uncertainties (1σ) to zircon (U-Th)/He data from a suite of samples previously dated in the CU TRaIL for combined F$_T$ and R$_{FT}$.**

| Sample and aliquot[a] | Geo.[b] | Max. Width[c] (µm) | combined F$_T$ | | | | | | | R$_{FT}$ | | | |
| | | | F$_{T, 2D}$ | F$_{T, GCM}$ | | | | | | R$_{FT,2D}$ | R$_{FT,GCM}$ | | |
| | | | F$_{T, 2D}$[d] | F$_{T, GCM}$[e] | ± TAU[f] | ± TAU[g] (%) | ± TAU + geom[h] | ± TAU + geom[i] (%) | | R$_{FT,2D}$[j] (µm) | R$_{FT, GCM}$[k] (µm) | ±[l] (µm) | ±[m] (%) |
| **RGD17-21** | | | | | | | | | | | | | |
| z02 | Tet. | 119.5 | 0.85 | 0.83 | 0.01 | 1% | 0.02 | 2% | | 79 | 73 | 5.8 | 8% |
| z03 | Tet. | 118.6 | 0.85 | 0.82 | 0.01 | 1% | 0.02 | 2% | | 79 | 73 | 5.8 | 8% |
| **PP4** | | | | | | | | | | | | | |
| z01 | Tet. | 90.5 | 0.77 | 0.74 | 0.02 | 2% | 0.03 | 3% | | 52 | 48 | 3.8 | 8% |
| z02 | Tet. | 108.5 | 0.84 | 0.82 | 0.01 | 1% | 0.02 | 2% | | 74 | 68 | 5.5 | 8% |
| z03 | Tet. | 121.4 | 0.84 | 0.81 | 0.01 | 1% | 0.02 | 2% | | 75 | 69 | 5.5 | 8% |
| **CA8** | | | | | | | | | | | | | |
| z01 | Tet. | 118.3 | 0.81 | 0.79 | 0.01 | 2% | 0.02 | 2% | | 62 | 57 | 4.6 | 8% |
| z02 | Tet. | 138.2 | 0.85 | 0.83 | 0.01 | 1% | 0.02 | 2% | | 79 | 73 | 5.8 | 8% |
| z03 | Tet. | 140.3 | 0.82 | 0.80 | 0.02 | 2% | 0.02 | 3% | | 66 | 61 | 4.9 | 8% |
| z04 | Ellip. | 85.1 | 0.76 | 0.76 | 0.02 | 3% | 0.03 | 4% | | 49 | 48 | 3.8 | 8% |
| **BP19-14** | | | | | | | | | | | | | |
| z01 | Tet. | 116.0 | 0.84 | 0.81 | 0.03 | 4% | 0.03 | 4% | | 74 | 68 | 5.4 | 8% |
| z02 | Tet. | 78.0 | 0.76 | 0.73 | 0.05 | 7% | 0.06 | 8% | | 49 | 45 | 3.6 | 8% |
| z03 | Ellip. | 140.0 | 0.80 | 0.80 | 0.01 | 1% | 0.02 | 3% | | 59 | 58 | 4.6 | 8% |
| z04 | Tet. | 50.0 | 0.65 | 0.63 | 0.01 | 2% | 0.02 | 3% | | 33 | 30 | 2.4 | 8% |
| z05 | Tet. | 104.0 | 0.85 | 0.82 | 0.06 | 8% | 0.06 | 8% | | 78 | 72 | 5.8 | 8% |
| z06 | Ellip. | 139.0 | 0.86 | 0.86 | 0.05 | 6% | 0.06 | 7% | | 84 | 83 | 6.6 | 8% |
| z07 | Tet. | 141.0 | 0.84 | 0.82 | 0.01 | 2% | 0.02 | 2% | | 74 | 68 | 5.5 | 8% |
| z08 | Ellip. | 139.0 | 0.85 | 0.85 | 0.04 | 4% | 0.04 | 5% | | 79 | 77 | 6.2 | 8% |
| z09 | Tet. | 41.0 | 0.61 | 0.59 | 0.01 | 2% | 0.02 | 3% | | 29 | 27 | 2.2 | 8% |
| z10 | Tet. | 81.0 | 0.76 | 0.73 | 0.01 | 1% | 0.02 | 3% | | 49 | 45 | 3.6 | 8% |
| z11 | Tet. | 149.0 | 0.82 | 0.80 | 0.02 | 3% | 0.03 | 3% | | 65 | 60 | 4.8 | 8% |
| **FCT** | | | | | | | | | | | | | |
| z36 | Tet. | 104.3 | 0.83 | 0.80 | 0.01 | 1% | 0.02 | 2% | | 67 | 61 | 4.9 | 8% |
| z37 | Tet. | 69.7 | 0.76 | 0.73 | 0.01 | 1% | 0.02 | 3% | | 46 | 43 | 3.4 | 8% |
| z38 | Tet. | 134.7 | 0.84 | 0.82 | 0.01 | 1% | 0.02 | 2% | | 73 | 67 | 5.4 | 8% |
| z39 | Tet. | 63.8 | 0.73 | 0.71 | 0.01 | 1% | 0.02 | 3% | | 42 | 38 | 3.1 | 8% |
| z40 | Tet. | 86.7 | 0.8 | 0.78 | 0.01 | 1% | 0.02 | 3% | | 59 | 54 | 4.3 | 8% |
| z41 | Tet. | 110.1 | 0.82 | 0.80 | 0.01 | 1% | 0.02 | 2% | | 66 | 60 | 4.8 | 8% |
| z42 | Tet. | 132.6 | 0.84 | 0.81 | 0.01 | 1% | 0.02 | 2% | | 72 | 67 | 5.3 | 8% |
| z43 | Tet. | 108.4 | 0.82 | 0.79 | 0.01 | 1% | 0.02 | 2% | | 64 | 58 | 4.7 | 8% |

**All uncertainties reported at the 1σ level.**



All calculations done assuming $F_T$ uncertainties are fully correlated (r = 1).

[a] All RGD17-21, PP4, CA8, and BP19-14 data are published in McGrew & Metcalf (2000), Havranek and Flowers (2022), Basler et al. (2021) and Peak et al. (2023), respectively.

[b] Geometry is defined as described in Figure 3 of Ketcham et al. (2011). All GEM A and B grains are tetragonal (tet.) ("tetrahedral prism" of Ketcham et al. (2011)) and all GEM C grains are ellipsoidal (ellip.).

[c] Maximum width is measured perpendicular to the length/c-axis.

[d] $F_{T,2D}$ is the combined alpha-ejection correction for the crystal calculated from the 2D parent isotope-specific $F_T$ corrections, the proportion of U and Th contributing to the $^4$He production, and assuming homogeneous parent isotope distributions using equation A4 in Cooperdock et al. (2019). The parent isotope-specific alpha-ejection corrections were computed assuming the reported grain geometry in this table and the equations and alpha-stopping distances in Ketcham et al. (2011).

[e] $F_{T,GCM}$ is computed the same as $F_{T,2D}$, but uses isotope-specific $F_{T,GCM}$ values corrected by applying the correction factors in Table 2 based on grain geometry and size.

[f] The 1σ TAU on $F_{T,GCM}$. This calculation uses 0% geometric uncertainty.

[g] The 1σ total analytical percent uncertainty on $F_{T,GCM}$.

[h] The 1σ TAU + geometric uncertainty. This uncertainty includes the total analytical uncertainty and uses the parent isotope-specific $F_{T,GCM}$ uncertainties assigned based on grain geometry and size (Table 2).

[i] The 1σ total analytical + geometric percent uncertainty on $F_{T,GCM}$.

[j] $R_{FT,2D}$ is the radius of a sphere with an equivalent alpha-ejection correction as the grain, calculated using the uncorrected parent isotope-specific $F_T$ values in equation A6 in Cooperdock et al. (2019).

[k] $R_{FT,GCM}$ is computed from $R_{FT,2D}$ by multiplying $R_{FT,2D}$ by the correction factor in Table 2 based on grain geometry.

[l] The 1σ uncertainty on $R_{FT,GCM}$ is assigned based on grain geometry (Table 2).

[m] The 1σ percent uncertainty on $R_{FT,2D}$.





**Table D3. Results of applying geometric corrections and uncertainties (1σ) to zircon (U-Th)/He data from a suite of samples previously dated in the CU TRaIL for corrected zircon (U-Th)/He date.**

| Sample and aliquot[a] | Geo.[b] | Max. Width[c] (μm) | Corrected zircon (U-Th)/He date | | | | | | | |
| | | | Date2D | | | DateGCM | | | | |
| | | | Date2D[d] (Ma) | ± TAU[e] (Ma) | ± TAU[f] (%) | DateGCM[g] (Ma) | ± TAU[h] (Ma) | ± TAU[i] (%) | ± TAU + geom[j] (Ma) | ± TAU + geom[k] (%) |
|---|---|---|---|---|---|---|---|---|---|---|
| **RGD17-21** | | | | | | | | | | |
| z02 | Tet. | 119.5 | 20.1 | 0.7 | 3% | 20.7 | 0.7 | 3% | 0.8 | 4% |
| z03 | Tet. | 118.6 | 16.1 | 0.3 | 2% | 16.6 | 0.4 | 2% | 0.5 | 3% |
| **PP4** | | | | | | | | | | |
| z01 | Tet. | 90.5 | 763.5 | 22.5 | 3% | 785.5 | 23.1 | 3% | 33.3 | 4% |
| z03 | Tet. | 108.5 | 342.0 | 6.8 | 2% | 352.3 | 7.0 | 2% | 10.1 | 3% |
| z03 | Tet. | 121.4 | 173.4 | 3.4 | 2% | 178.6 | 3.5 | 2% | 5.1 | 3% |
| **CA8** | | | | | | | | | | |
| z01 | Tet. | 118.3 | 186.0 | 5.2 | 3% | 191.7 | 5.4 | 3% | 6.7 | 3% |
| z02 | Tet. | 138.2 | 173.8 | 3.2 | 2% | 179.1 | 3.3 | 2% | 5.0 | 3% |
| z03 | Tet. | 140.3 | 173.9 | 4.5 | 3% | 179.2 | 4.6 | 3% | 6.0 | 3% |
| z04 | Ellip. | 85.1 | 121.1 | 3.6 | 3% | 121.1 | 3.6 | 3% | 5.2 | 4% |
| **BP19-14** | | | | | | | | | | |
| z01 | Tet. | 116.0 | 533.7 | 15.2 | 3% | 549.4 | 15.6 | 3% | 19.0 | 3% |
| z02 | Tet. | 78.0 | 578.1 | 24.0 | 4% | 595.0 | 24.7 | 4% | 30.2 | 5% |
| z03 | Ellip. | 140.0 | 616.4 | 16.4 | 3% | 616.4 | 16.4 | 3% | 24.3 | 4% |
| z04 | Tet. | 50.0 | 561.6 | 25.4 | 5% | 578.1 | 26.1 | 5% | 31.0 | 5% |
| z05 | Tet. | 104.0 | 561.1 | 21.7 | 4% | 577.6 | 22.3 | 4% | 25.0 | 4% |
| z06 | Ellip. | 139.0 | 207.5 | 13.3 | 6% | 207.5 | 13.3 | 6% | 14.7 | 7% |
| z07 | Tet. | 141.0 | 356.1 | 11.0 | 3% | 366.7 | 11.3 | 3% | 13.6 | 4% |
| z08 | Ellip. | 139.0 | 726.7 | 37.1 | 5% | 726.7 | 37.1 | 5% | 42.7 | 6% |
| z09 | Tet. | 41.0 | 609.9 | 23.8 | 4% | 627.7 | 24.5 | 4% | 30.5 | 5% |
| z10 | Tet. | 81.0 | 607.8 | 13.3 | 2% | 625.6 | 13.7 | 2% | 23.0 | 4% |
| z11 | Tet. | 149.0 | 482.2 | 15.8 | 3% | 496.5 | 16.3 | 3% | 19.0 | 4% |
| **FCT** | | | | | | | | | | |
| z36 | Tet. | 104.3 | 27.1 | 0.8 | 3% | 28.0 | 0.8 | 3% | 1.0 | 4% |
| z37 | Tet. | 69.7 | 26.3 | 0.7 | 3% | 27.1 | 0.7 | 3% | 1.1 | 4% |
| z38 | Tet. | 134.7 | 26.7 | 0.6 | 2% | 27.5 | 0.6 | 2% | 0.9 | 3% |
| z39 | Tet. | 63.8 | 28.2 | 0.6 | 2% | 29.1 | 0.6 | 2% | 1.1 | 4% |
| z40 | Tet. | 86.7 | 29.8 | 0.5 | 2% | 30.7 | 0.5 | 2% | 1.2 | 4% |
| z41 | Tet. | 110.1 | 25.6 | 1.0 | 4% | 26.4 | 1.0 | 4% | 1.2 | 4% |
| z42 | Tet. | 132.6 | 32.1 | 0.7 | 2% | 33.1 | 0.7 | 2% | 1.0 | 3% |
| z43 | Tet. | 108.4 | 27.4 | 0.7 | 2% | 28.3 | 0.7 | 2% | 0.9 | 3% |

All uncertainties reported at the 1σ level.

All calculations done assuming $F_T$ uncertainties are fully correlated (r = 1).

[a] All RGD17-21, PP4, CA8, and BP19-14 data are published in McGrew & Metcalf (2000), Havranek and Flowers (2022), Basler et al. (2021) and Peak et al. (2023), respectively.

[b] Geometry is defined as described in Figure 3 of Ketcham et al. (2011). All GEM A and B grains are tetragonal (tet.) ("tetrahedral

prism" of Ketcham et al. (2011)) and all GEM C grains are ellipsoidal (ellip.).

[c] Maximum width is measured perpendicular to the length/c-axis.

[d] The corrected (U-Th)/He date2D is calculated iteratively using the absolute values of He, U, Th, Sm, the isotope-specific $F_{T,2D}$ values, and equation 34 in Ketcham et al. (2011) assuming secular equilibrium.

[e] The 1σ TAU uncertainty on date2D includes the propagated total analytical uncertainties on the U, Th, Sm and He measurements.

Uncertainty propagation done using HeCalc (Martin et al., 2023).

[f] The 1σ total analytical percent uncertainty on date2D.



[g] The corrected (U-Th)/He date$_{GCM}$ is computed the same as date$_{2D}$, but uses the isotope-specific $F_{T,GCM}$ values corrected by applying the correction factors in Table 2 based on grain geometry and size.

[h] The 1σ TAU uncertainty on the corrected (U-Th)/He date$_{GCM}$ includes the propagated total analytical uncertainties on the U, Th, Sm, He measurements. This calculation uses 0% geometric uncertainty. Uncertainty propagation done using HeCalc (Martin et al., 2023).

[i] The 1σ total analytical percent uncertainty on the corrected (U-Th)/He date$_{GCM}$.

[j] The 1σ total analytical + geometric uncertainty on the corrected (U-Th)/He date$_{GCM}$. This uncertainty includes the propagated total analytical uncertainties on the U, Th, Sm, He measurements and uses the parent isotope-specific $F_{T,GCM}$ uncertainties 945     assigned based on grain geometry and size (Table 2).

[k] The 1σ total analytical + geometric percent uncertainty on the corrected (U-Th)/He date$_{GCM}$.

**Code and Data Availability**

Raw data and code used to produce the corrections, uncertainties, and figures are stored through the Open Science Framework doi.org/10.17605/OSF.IO/QRK4J. All analyses and plots were done in R (R 950     Core Team, 2023; Wickham et al., 2019).

**Author contributions**

RMF and JRM conceptualized the project; MB and SDZ curated the data; MB, SDZ and JRM performed the formal data analysis; RMF, JRM, and SDZ acquired funding; MB performed the investigation; JRM, SDZ, MB, and RMF developed the methodology; RMF provided supervision; MB 955     and SDZ performed the validation; SDZ and MB did the data visualizations; MB, SDZ, and RMF wrote the original draft; RMF, SDZ, MB, and JRM reviewed and edited the paper.

**Competing Interests**

The authors declare they have no conflict of interest.

**Acknowledgements**

We would like to thank Brian Mahoney and Jacky Baughman for samples. We are grateful to Adrian Gestos for analyzing our samples on the nano-CT during the pandemic and his continuing guidance and assistance.

**Financial Support**

This research has been supported by the National Science Foundation Graduate Research Fellowship 965     (grant no. DGE-1650115) to Spencer D. Zeigler. Funding for the Zeiss Xradia Versa X-ray microscope



was provided by the National Science Foundation (grant no. CMMI-1726864). Partial funding for analytical costs was provided by The Beverly Sears Graduate Student Grant Fund at CU Boulder to Spencer D. Zeigler.