# Peer review of "The Geometric Correction Method for zircon (U-Th)/He chronology: correcting systematic error and assigning uncertainties to alpha-ejection corrections and eU concentrations"

_EGUsphere, 2023_

## Author Comment (AC1)

| Sample Name | Unit and Lithology | Sample Age | Locality | Latitude (°N) | Longitude (°W) | GEM Categories | N[a] | Additional Geochronologic and Thermochronologic Data |
|---|---|---|---|---|---|---|---|---|
| FCT | Fish Canyon Tuff, Dacite | Oligocene | San Juan Mountains, Colorado, USA | 37.756 | 106.934 | A | 23 | Zircon U-Pb 28.172 ± 0.028 Ma (2sigma) (Schmitz and Bowring, 2001); ZHe 28.7 ± 0.4 Ma (1sigma) (Gleadow et al., 2015) |
| RGD17-21 | Harrison Pass Pluton, Granodiorite | Eocene | Ruby Mountains, Nevada, USA | 40.326 | 115.510 | A, B | 23 | Zircon U-Pb ca. 36 Ma (Wright and Snoke, 1993); ZHe 20-16 Ma[b] (McGrew & Metcalf., 2000) |
| CA8 | Potomac terrane, Quartzofeldspathic schist | Precambrian | Appalachian Mountains, Virginia, USA | 37.984 | 78.311 | A, B | 27 | ZHe 186-121 Ma[b] (Basler et al., 2021) |
| PP4 | Pikes Peak Batholith, Syenogranite | Proterozoic | Pikes Peak, Colorado, USA | 38.842 | 105.025 | A, B | 20 | Hornblende & Biotite $^{40}$Ar/$^{39}$Ar 1.08-1.07 Ga (Unruh, 1995); ZHe 115- 773 Ma[b] (Havranek and Flowers, 2022) |
| CP06-70 | 245-Mile Complex, Granodiorite | Proterozoic | Grand Canyon, AZ, USA | 35.843 | 113.599 | A, B, C | 39 | Zircon U-Pb ca. 1700 Ma (Hawkins et al.,1996); ZHe 560-96 Ma[b] (2sigma) (Peak et al., 2021) |
| 01-OE-38 | Migmatitic Gneiss | Archean | Superior craton, Canada | 47.270 | 84.560 | A, B | 24 | Zircon U–Pb 2720–2680 Ma (Hoffman, 1989); AHe 275-34 Ma[b] (Sturrock et al., 2024) |
| 56JBM14 | Río de los Patos Frm., medium-grained tuffaceous sandstone | Paleogene | Manantiales Basin, Argentina | -32.050 | 69.750 | B | 10 | Zircon U-Pb 38.68 ± 0.21 (2sigma) (Suriano et al., 2023) |
| CP06-14 | Coconino Sandstone | Permian | Colorado Plateau, AZ, USA | 34.300 | 110.901 | C | 28 | No geochronologic data for this sample |
| CP06-15 | Esplanade Sandstone | Permian | Colorado Plateau, AZ, USA | 34.298 | 110.906 | B, C | 29 | No geochronologic data for this sample |

---

## Author Response (AR1)

Dear Dr. Tremblay,

Thank you for your prompt decision on our manuscript and your additional comments and suggestions. This is an excellent idea to include the zircon photomicrographs in our data repository so that anyone can use the data to develop geometric corrections and uncertainties for any approach – we simply had not thought of this. We added the photomicrographs to the repository and updated the data availability statement (L1087). We have also updated the README in our "scripts" folder on the data repository; the file you noticed was 'missing' (comp-with-other-studies.R) was never supposed to be uploaded—the description of the script was accidentally left in the README. The script was not used in this manuscript.

Additionally, during editing we noticed a mistake in our Appendix A and changed the ellipsoid V (A5) and SA (A6) equations to reflect the use of the semi axis widths in Ketcham et al. (2011) (L875-890). An "in review" manuscript we cited in Table 1 was published and that citation was updated (L125; L830). Over the course of addressing reviews, we added a few citations (L702, L735, L753, L784, L817). Finally, on L1104 we added a thank you to the reviewers.

Below you will find a point-by-point response to each of the reviewer's comments. Our responses are in bold, and the specific changes are in dark blue text indented with a dash. The specific text we added to address the reviews is copied from the manuscript, bulleted, and italicized.

Thank you,

Spencer Zeigler et al.

RC1 (Florian Hoffman)

**We are grateful for Florian's kind and thorough review. We agree with his suggestions, made the associated modifications to the manuscript, and include our responses in bold below.**

Detailed Comments:

Line 78: Do the numbers for the resolution (0.84-0.92 μm) refer to the voxel size or the smallest possible distance between two objects that can be resolved?

**They refer to the voxel size. We have added language to clarify this point in the introduction.**

- See L89: "(sub-1 μm; voxel sizes of 0.84-0.92 μm)"

Table 1: The formatting makes it hard to read, especially the second column. It's not immediately apparent which lines belong to which sample. I suggest adding horizontal lines or additional space to separate the rows from each other.

**We have made changes to this table to increase legibility. Further changes to legibility will be made during the proofing process if necessary (see attachment).**

- See Table 1 (L125), we added horizontal lines to separate samples and added space to increase legibility. Additionally, we edited the GEM Categories column to reflect the use of the GEM chart in the main text.

Line 399: Insert spaces between number and unit to make it consistent with the rest of the text: "100μm" --> "100 μm". Also change elsewhere (Figure C1, Line 138, etc.).

**Thank you. We have made this change throughout the text and in Figures: 2, 7, and C1.**

- Figure 2: Added a space between # and μm in the legend. (L170)
- Figure 7: Added a space between # and μm in the legend. (L344)
- Figure C1: Added a space between # and μm in the legend. (L970)
- Main text: L89, L164, L320, L321, L351, L455

Lines 434-441: This might be beyond the scope of this paper to discuss, but I'm wondering how the 2D and 3D volume-derived masses would compare to ICP-MS-derived Zr-based masses for zircon grains. Some labs measure Zr routinely and use that for calculating grain mass (e.g. Guenthner et al., 2016, G3). Using those two approaches concurrently could be used to derive the average density of the zircon grains, which correlates to the amount of crystal damage. The difference in density between pristine and highly metamict zircons of around 16% (as mentioned in Line 530) should be resolvable given the uncertainties mentioned in this manuscript.

**This is an interesting idea, but we agree that this is beyond the scope of this paper. We have added mention of the stoichiometric approach described by Guenthner et al. (2016) for zircon mass determination and how those outcomes compare with our results. As you point out, both our approach and the stoichiometric approach require assuming the density of zircon, which is another source of uncertainty.**

- We added a discussion of the stoichiometric approach by Guenther et al. (2016) to L427-430.
  - *"Our results agree with those of a study that compared tetragonal zircon masses calculated by traditional microscopy measurements with those measured by isotope dilution ICP-MS of Zr and assuming zircon stoichiometry, which found that 2D values overestimate the stoichiometric results with an average percent difference of 25% between the two methods (Guenthner et al., 2016)".*

RC 2 (Anonymous)

**We thank the anonymous reviewer for their detailed review which will improve our manuscript. We have included replies (in bold) to each of their points and suggestions below.**

In section 3.2, the authors assumed a zircon Th/U ratio of 0.87 and no Sm contribution owing to a lack of parent isotope measurements. Is it possible to perform some kind of supplementary analysis or offer more reasoning to demonstrate that the use of the assumed values (as opposed to sample-specific parent isotope measurements) would not lead to a significant difference in a series of calculated results presented later, nor the major conclusion?

**The value for the Th/U ratio comes from the average of 736 zircon analyses in the CU Boulder TRaIL. We used this value mainly for illustrative purposes (i.e., to calculate the combined FT and RFT shown in Figure 7). However, the Th/U ratio is also used to calculate the correction and uncertainty for the combined FT and RFT. But, because the ratio is used in the calculation of both the 3D and 2D values for each parameter, the actual value of the correction and uncertainty are not impacted by the choice of Th/U (i.e., the choice of Th/U "cancels out" during the regression). The actual measured elemental concentrations are used to calculate combined FT and RFT when the corrections and uncertainties are applied to real grains. We have added language to clarify this point.**

- *See L260-261*
  - *"The assumed Th/U ratio does not impact the value of the corrections or uncertainty and is only used for illustrative purposes."*

The collapsed clarity dimension. I do enjoy reading the discussions of how zircon clarify is related to eU, which is a critical proxy for radiation damage accumulation and annealing, and the density of zircon. However, I am a little confused about the way the authors delivered their reasonings about abandoning the clarity dimension in the GEM (section 2.4, around lines 160-168). My understanding is that the manuscript is centered on the assessment of error and uncertainty in zircon dimensions, therefore one should be able to exclude the contribution of clarity in terms of the role of varying eU before performing analyses. However, as the author pointed out, zircon clarity is also related to its density, so I think this part of the role of zircon clarity should only be either retained or abandoned after showcasing the analyses presented later on. Therefore, does it make more sense to not abandon this dimension at this point of the manuscript? Moving on, it seems that the authors use 1, 2, or 3 as a numerical index for grain clarity if I understood correctly. Could the clarity be treated as a continuum (through some methods like a grayscale image?) and would a different conclusion on the importance of grain clarity?

**Thanks for this comment. We wanted to emphasize that noting clarity when picking is important given its relationship to radiation damage and density. But we see your point and have downplayed the discussion of the clarity axis by making edits to section 2.4 and moving the discussion about the GEM design to Appendix B.**

**We use 1, 2, and 3 as a numerical index for grain clarity due to the unreliability between analysts categorizing clarity into any finer of bins. We favor treating clarity as a qualitative proxy for radiation and therefore retain that axis on the GEM presented in Appendix B. For analysts who prefer more categories to reflect intrasample variation in visual metamictization, we refer the readers to Armstrong et al. (2024) (and have included this citation in our manuscript).**

- We edited the discussion to be less focused on clarity and included the Armstrong et al. reference in the main text (see L190-210)
  - *"Zircon grain clarity is known to correlate with radiation damage (e.g., Ault et al. 2018; Armstrong et al., 2024), which influences He retentivity and zircon density, making grain clarity useful information to record during grain selection. We include further discussion of grain clarity and a two-axis zircon GEM in Appendix B but we do not discuss zircon clarity further in the main text as this parameter does not impact the geometric corrections and uncertainties (Table C2)."*

- We moved the clarity discussion to Appendix B for interested readers (L920-935)
    - *"The zircon-GEM was initially designed with two axes: a "geometric classification" x-axis and a "clarity index" y-axis (Fig. B1). Zircon grain clarity was initially considered because this characteristic correlates with radiation damage (e.g., Ault et al., 2018; Armstrong et al., 2024), which influences zircon He retentivity (and therefore the ZHe date) and zircon density (and therefore the estimated mass and eU values). Grain clarity thus can be useful information to record during grain selection and is retained in the two-axis zircon GEM (Fig. B1). We chose to use a small number of discrete categories for grain clarity due to the difficulty and inconsistency between analysts of categorizing grains into even finer categories. However, zircon clarity does not impact geometric corrections and uncertainties (Table C2), so the zircon GEM was collapsed to a single "geometric classification" axis as shown in the main text (Fig. 3)."*
- We made edits to the "GEM" column in Table 1 to reflect the use of the "condensed GEM" in the main text (L125).

Line 78: Is there a reason that the resolution of the CT data is presented as a range? I am not sure if additional information is beneficial though. Regardless, at this scale, would it make sense to simplify it as ~1 μm for ease of reading?

**We originally present the CT resolution as a range because the actual resolutions measured by the CT varied as a function of scan time, magnification, voltage, and power (see Appendix Table B1). We agree that for readability it would be simpler to report the resolution as "sub-1 μm" and have made this change throughout the text (3 instances).**

- L89
- L220
- L277

Line 100: Table 1 might look clearer if (1) more space is allowed between rows separating different sample suites or (2) adapting a similar formatting to tables 2-4.

**We agree and have made changes to improve readability. Further changes to legibility will be made during the proofing process if necessary (see attachment).**

- See Table 1 (L125). We added horizontal lines to separate samples, added space to increase legibility, and edited the GEM Categories column to reflect the use of the GEM chart in the main text.

Line 138: additional space between number and unit.

**We have made this change.**

- Figure 2: Added a space between # and μm in the legend. (L170)
- Figure 7: Added a space between # and μm in the legend. (L344)
- Figure C1: Added a space between # and μm in the legend. (L970)
- Main text: L89, L164, L320, L321, L351, L455

Line 179-181: Minor comment. The first sentence of section 3.1 is a bit repetitive. However, this starting sentence, whether or not in a revised form, seems to be a more explicit way to introduce the first-order goal of this study if placed in the Intro section.

**Thank you for your comment..**

- We made minor edits to this sentence in Section 3.1 (L217).
  - *"To determine corrections for systematic error and appropriate uncertainties arising from traditional "2D" microscopy measurements we compare 2D geometric parameters with "3D" geometric parameters acquired via CT."*

Line 266-268: Would it be better if the definition of the corrections for systematic error could be more explicitly defined here? (e.g., incorporate details from Table 2 footnotes).

**We define the systematic error in line 53 [54] and will add a clarifying note from the Table 2 footnotes to L266-268.**

- Added a clarifying note to L313
  - *"To determine the corrections for systematic error (e.g., the slope of the regression) we…"*

RC3 (Willy Guenthner)

**We thank Willy for his helpful review that will improve our manuscript! We address each of his points and suggestions below, with our replies in bold.**

- I am wondering how the Ft error would be different if a different set of 2D geometry equations were used. Both Hourigan et al. (2005, https://doi.org/10.1016/j.gca.2005.01.024) and Reiners et al. (2005, American Journal of Science v. 305, p. 259-311) derive a set of Ft equations that take into the account the specific dimensions of the two pyramidal terminations in a given zircon grain. Specifically, see equations 1-3 in the Reiners et al. (2005) article. The Ketcham et al. (2011) equations do take the pyramid heights into account, but only as a uniform approximation, whereas the Hourigan et al. equations, further derived by Reiners et al., are grain specific. My question then is, is the magnitude of the errors that result from the 2D and 3D comparison of volume and Ft in figure 7 somewhat mitigated by using equations that incorporate metrics for the terminations? The authors seem to suggest as much at line 390. I would encourage the authors to examine this by measuring tip heights directly from their collected images. This is significant in so far as the authors are proposing a general correction that could be applied to previously published datasets, and I know from experience that all zircon (U-Th)/He data sets generated at the University of Arizona (of which there are a bunch in the published literature at this point) use the Reiners et al. (2005) correction and not the Ketcham et al. (2011).

**Thank you for this excellent point. We will add clarifying language around our decision to use the Ketcham (2011) equations over alternative approaches. We will also emphasize that the use of the Reiners et al. (2005) or Hourigan et al. (2005) geometry equations do not preclude the application of our corrections and uncertainties to previously published data. The mean length and width can be**

**derived from measurements of the trunk and tip height and can be incorporated into the Ketcham et al. (2011) approach. We will add language to emphasize this fact.**

**We agree that it would be interesting to see how these different 2D methods impact the corrections and we would be happy to discuss sharing our grain images if you would like to collaborate on a technical note!**

- We added a justification of using the Ketcham et al. calculations (L251-252) and explanation of how the corrections can be applied to all published datasets regardless of initial method used to calculated geometry (L663-668).
    - *"We chose to use the Ketcham et al. (2011) method because the equations can incorporate updated stopping distances in the future without need for reformulation."*
    - *"This study uses the Ketcham et al. (2011) equations for computing the geometric parameters. Other methods for computing the volume and surface area of zircon crystals are available that incorporate measurements of the pyramidal termination height (e.g., Reiners et al., 2005; Hourigan et al., 2005), but this does not preclude the application of our corrections and uncertainties to these datasets because the mean length and width of the zircon crystals can be derived from the reported measurements and incorporated into the Ketcham et al. (2011) calculations if desired."*
- At the suggestion of the AE, have now placed all photomicrographs in the data repository and added an update to the data availability statement (L1087). This will enable anyone to use the data to develop geometric corrections and uncertainties for any approach, including for the Reiners et al. (2005) and Hourigan et al. (2005) equations.
    - *"Raw data, photomicrographs of all zircon crystals used in this study, and code used to produce the corrections, uncertainties, and figures are stored through the Open Science Framework doi.org/10.17605/OSF.IO/QRK4J."*

- There is little mention in the current version of the manuscript concerning U and Th zonation and its influence on the Ft correction. The authors have defined the scope of the current work to focus on corrections to potential systematic error introduced by relying on a 2D geometric approximation only, and this is fine and appropriate. A full consideration of this influence plus the zonation effect would likely be a separate study. However, again, given that the authors propose a general-use error assignment for all zircon grains with only 2D measurements, my concern is that this correction could still give false confidence in what the actual, true Ft error correction should be. For example, common scenarios of U and Th zonation in zircon can lead to Ft correction errors of ~30 % (Hourigan et al., 2005), which is far greater than the errors discussed here. How do the authors consider their findings in comparison to those of Hourigan et al. (2005) and their recommendations for Ft error correction? Zonation is admittedly a pernicious issue, and so there might not be a great answer to this question without resorting to time-intensive in situ approaches, but the authors do need to elaborate more on the error discrepancies here.

**We acknowledge that zonation can have a large influence on the magnitude of corrections and uncertainties. We stated at multiple places in the original paper that our corrections do not account for zonation, but to combat the real issue of "false confidence" we will add language to emphasize this more strongly in the introduction.**

**We will also explain in the introduction that our strategy in this study and some of our other work has been to compartmentalize and characterize different sources of uncertainty. This study is aimed at characterizing the geometric uncertainties associated with ZHe dates. Additional work could characterize the uncertainty arising from zonation, and then that uncertainty could additionally be propagated into the ZHe data. We see this paper as part of an ongoing effort in the thermochronology community to carefully characterize the different uncertainty components in (U-Th)/He dates.**

- We added language to the introduction to emphasize up front in the manuscript that our corrections and uncertainties do not take zonation into account at L73-75; L224-225
    - *"This work is focused on characterizing the uncertainty and inaccuracy from assumptions about grain geometry only, and does not account for additional contributions from parent isotope zonation (e.g., Farley et al., 1996; Hourigan et al., 2005), grain abrasion (e.g., Rahl et al., 2003), grain breakage (He and Reiners, 2022), and zircon density (e.g., Holland and Gottfried, 1955), which have potential to be accounted for separately."*
    - *"The uncertainties presented here include only those associated with grain geometry and not those due to parent isotope zonation, grain abrasion, or crystal breakage."*
- We added language (and citations) in the introduction and in section 5.3 to more clearly explain that this work is a part of an ongoing effort to compartmentalize and characterize different source of uncertainty (L97-100; L588-590)
    - *"This study is part of ongoing efforts by the thermochronology community to carefully quantify and account for the different sources of uncertainty in (U-Th)/He data (e.g., Martin et al., 2023; Guenthner et al., 2016; Cooperdock et al., 2019; Zeigler et al., 2023; Flowers et al., 2022a and references therein), all of which could then be propagated into the reported uncertainties of (U-Th)/He results."*
    - *"As additional sources of uncertainty are characterized, these too can be propagated into the uncertainties on (U-Th)/He data."*

How do the results here compare with the Zr stoichiometric approach of Guenthner et al. (2016)? A comparison of Figure 4a in Guenthner et al. (2016) and Figure 7a seems to suggest that there might be good first-order agreement between the 3D approach and the stoichiometric approach. Have these zircon grains already been dissolved? A useful follow-up would be a direct comparison among the 3D, 2D, and stoichiometric approach. This is likely outside the scope of the current study, but some additional discussion along these lines is warranted in the current text.

**These grains have not been dissolved. This is a good idea and we also have previously discussed such a follow up study as part of setting up the stoichiometric method in our lab. Our lab has explored the stoichiometric approach several times since 2012 but have not felt comfortable implementing it without a way to independently "ground truth" the results. We agree that these zircon grains provide the opportunity to carry out a rigorous comparative study. We will also add mention of how the masses and volumes determined in this study compare with the results of Guenthner et al. (2016).**

- L426-430: We added a comparison of our results with those from Guenther et al (2016).

- o *"Our results agree with those of a study that compared tetragonal zircon masses calculated by traditional microscopy measurements with those measured by isotope dilution ICP-MS of Zr and assuming zircon stoichiometry, which found that 2D values overestimate the stoichiometric results with an average percent difference of 25% between the two methods (Guenthner et al., 2016)."*